# PHYBench: Holistic Evaluation of Physical Perception and Reasoning in Large Language Models

**Shi Qiu**[1,*], **Shaoyang Guo**[1,*], **Zhuo-Yang Song**[1,*], **Yunbo Sun**[1,*], **Zeyu Cai**[1,*], **Jiashen Wei**[1,*], **Tianyu Luo**[1,*], **Yixuan Yin**[1], **Haoxu Zhang**[1], **Yi Hu**[2], **Chenyang Wang**[1], **Chencheng Tang**[1], **Haoling Chang**[1], **Qi Liu**[1], **Ziheng Zhou**[1], **Tianyu Zhang**[1], **Jingtian Zhang**[1], **Zhangyi Liu**[1], **Minghao Li**[1], **Yuku Zhang**[1], **Boxuan Jing**[1], **Xianqi Yin**[1], **Yutong Ren**[1], **Zizhuo Fu**[2], **Jiaming Ji**[2], **Weike Wang**[1], **Xudong Tian**[1], **Anqi Lv**[1], **Laifu Man**[1], **Jianxiang Li**[1], **Feiyu Tao**[1], **Qihua Sun**[1], **Zhou Liang**[1], **Yushu Mu**[1], **Zhongxuan Li**[1], **Jing-Jun Zhang**[1], **Shutao Zhang**[1], **Xiaotian Li**[1], **Xingqi Xia**[1], **Jiawei Lin**[1], **Zheyu Shen**[1], **Jiahang Chen**[1], **Qiuhao Xiong**[1], **Binran Wang**[1], **Fengyuan Wang**[1], **Ziyang Ni**[1], **Bohan Zhang**[5], **Fan Cui**[4], **Changkun Shao**[1], **Qing-Hong Cao**[1,†], **Ming-xing Luo**[3,†], **Yaodong Yang**[2], **Muhan Zhang**[2], and **Hua Xing Zhu**[1,†]

[1]School of Physics, Peking University
[2]Institute for Artificial Intelligence, Peking University
[3]Beijing Computational Science Research Center
[4]School of Integrated Circuits, Peking University
[5]Yuanpei College, Peking University

## Abstract

Current benchmarks for evaluating the reasoning capabilities of Large Language Models (LLMs) face significant limitations: task oversimplification, data contamination, and flawed evaluation items. These deficiencies necessitate more rigorous assessment methods. To address these limitations, we introduce PHYBench, a benchmark of 500 original physics problems ranging from high school to Physics Olympiad difficulty. PHYBench addresses data contamination through original content and employs a systematic curation pipeline to eliminate flawed items. Evaluations show that PHYBench activates more tokens and provides stronger differentiation between reasoning models compared to other baselines like AIME 2024, OlympiadBench and GPQA. Even the best-performing model, Gemini 2.5 Pro, achieves only 36.9% accuracy compared to human experts' 61.9%. To further enhance evaluation precision, we introduce the Expression Edit Distance (EED) Score for mathematical expression assessment, which improves sample efficiency by 204% over binary scoring. Moreover, PHYBench effectively elicits multi-step and multi-condition reasoning, providing a platform for examining models' reasoning robustness, preferences, and deficiencies. The benchmark results and dataset are publicly available at *https://www.phybench.cn/*.

## 1 Introduction

*"Benchmarks don't idolize or diminish models; they guide humanity and AI together toward AGI."*

Recent advances in reasoning models have significantly improved the reasoning capabilities of LLMs [6, 18, 23]. Evaluation frameworks such as MathArena [1] have demonstrated that frontier

---

* Equal Contribution.

†Corresponding author, correspondence to zhuhx@pku.edu.cn, qinghongcao@pku.edu.cn and mingxingluo@csrc.ac.cn

39th Conference on Neural Information Processing Systems (NeurIPS 2025) Track on Datasets and Benchmarks.

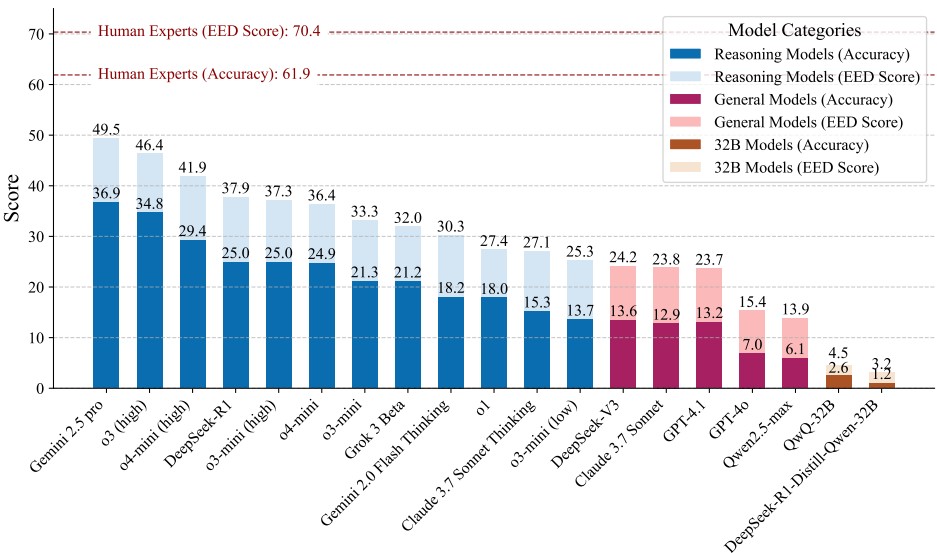

Figure 1: Model performance on PHYBench. We report accuracy and EED Score for both reasoning and general language models, averaged over all samples.

LLMs can already understand and answer problems at Olympiad Competition difficulty level. However, existing benchmarks may fail to accurately reflect and effectively distinguish between models due to three critical limitations: **(1) Oversimplified Reasoning Tasks.** State-of-the-art reasoning models exhibit performance saturation on traditional benchmarks. For example, DeepSeek-R1 [6] achieves an accuracy score of 97.3% on the MATH-500 dataset [13]. **(2) Potential Data Contamination.** Most existing datasets are constructed from publicly available materials that models may have encountered during pretraining. **(3) Lack of Rigorous Verification.** Many benchmarks [10, 13] include flawed questions or scoring criteria, which reduce models' instruction-following accuracy, introducing noise unrelated to actual reasoning performance. A more detailed discussion and illustrative examples for each of these limitations are provided in Appendix A.

To address these limitations, we introduce PHYBench, a challenging, human-curated benchmark designed to rigorously evaluate models' reasoning capabilities using physics problems. PHYBench covers diverse domains including mechanics, electromagnetism, thermodynamics, optics, modern physics and advanced physics. The questions span difficulty levels from high school physics to undergraduate coursework and Physics Olympiad problems. PHYBench consists **entirely of original problems** to eliminate data contamination and is designed to assess models' physical perception and robust reasoning capabilities. Based on this high-quality dataset, we propose the EED Score, an interpretable, fine-grained metric that measures the similarity between model-generated and reference expressions using tree edit distance. EED provides more nuanced and reliable scoring, improving sample efficiency by 204% on PHYBench.

We evaluate a wide range of LLMs on the PHYBench benchmark and additionally establish a human baseline by recruiting undergraduate students from Peking University, School of Physics to solve the same problems. The results indicate a clear performance gap: even the best-performing LLM, Gemini 2.5 Pro [25], achieved 36.9% accuracy, compared to the human baseline of 61.9% (detailed in Section 4). Compared to widely used benchmarks, PHYBench requires significantly more output tokens and yields lower model scores, highlighting its greater complexity and difficulty. PHYBench also provides stronger differentiation of reasoning abilities among models. In addition, our test-time scaling (TTS) [15, 28, 29] experiments show that PHYBench exhibits strong order-preservation under both pass@$k$ and majority voting settings. Further analysis reveals that many model errors originate from introducing incorrect conditions or equations during intermediate steps; models also exhibit a limited capacity to detect or correct these mistakes. Our key contributions are summarized as follows:

**A Challenging Physical Reasoning Benchmark.** We propose PHYBench, the first human-curated, high-quality benchmark designed to rigorously evaluate models' complex reasoning capabilities

using physics problems. PHYBench is constructed through a stringent curation pipeline to ensure that all problems are novel, correct, and reliably evaluable.

**A Fine-Grained Evaluation Metric.** We introduce EED Score, an interpretable, rule-based evaluation metric that measures similarity between model-generated and reference expressions by computing the edit distance over their tree structures. EED Score provides a continuous measure and robust assessment of solution correctness, and improves sample efficiency by 204% on PHYBench.

**An In-depth Analysis of LLM Reasoning.** Our analysis reveals a significant gap between LLMs and human experts in complex reasoning tasks. In particular, model errors arise from introducing incorrect conditions or equations in intermediate steps, and models lack the ability to detect or correct these mistakes, unlike the consistent self-checking behavior seen in human reasoning.

## 2   Related Work

**Reasoning Benchmarks.** As state-of-the-art models increasingly approach saturation on traditional benchmarks such as GSM-8K [4], Math-500 [13], and MMLU [4], marginal gains and potential overfitting have become notable concerns [6, 18]. Recent efforts aim to address this by introducing benchmarks that focus on frontier scientific knowledge, such as HLE [8], or on increased problem complexity, as in OlympiadBench [10] and AIME 2024 [11]. However, benchmarks in the former category emphasize knowledge coverage rather than reasoning, and thus fall outside the scope of reasoning-oriented evaluation. Benchmarks in the latter group often rely on publicly available problems, which lack originality and risk contamination due to prior exposure during model pretraining. To ensure reliable assessment, benchmarks based on original problems must undergo rigorous expert calibration to reduce ambiguity and ensure fairness. PHYBench addresses this gap by providing a fully original, human-curated dataset of 500 problems, specifically designed to evaluate complex reasoning in realistic physical contexts while avoiding data leakage and enabling precise evaluation.

**Evaluation Metrics for Complex Reasoning Tasks.** Traditional benchmarks often rely on multiple-choice or simple numerical answers, as in SuperGPQA [7] and MMLU [4]. These formats are easy to score but fail to reflect genuine reasoning, as answers may be chosen through elimination or pattern matching. Recent approaches have explored human evaluation or model-assisted scoring to assess reasoning processes in more detail. While human judgments offer the highest fidelity, they are costly and hard to scale. Model-assisted evaluation provides partial insight into intermediate reasoning steps but suffers from bias and instability, limiting its reliability. Some benchmarks, such as OlympiadBench [10] and AIME 2024, use expression or number-based binary scoring, which enforces answer format consistency but overlooks partial correctness. To address these limitations, we introduce **EED Score**, a symbolic expression-based metric built on SymPy [14] expression trees and extended tree edit distance. EED Score supports fine-grained comparison between model-generated and reference answers, enabling robust evaluation of reasoning quality beyond binary correctness.

## 3   The PHYBench Benchmark

### 3.1   Overview

Table 1: Comparison between PHYBench and other reasoning benchmarks. The Average Output Tokens and Average Accuracy are computed using DeepSeek-R1 [6].

| Dataset | Data Scale | Avg. Output Tokens | Avg. Accuracy | Scoring Type |
|---|---|---|---|---|
| MATH-500 [13] | 500 | 1857 | 97.3 | Binary |
| GPQA [24] | 448 | 6308 | 71.5 | Binary |
| OlympiadBench [10] | 8K | 5372 | 58.7 | Binary |
| AIME 2024 [11] | 30 | 7741 | 79.8 | Binary |
| **PHYBench (Ours)** | 500 | **10636** | 25.0 | **Detailed** |

PHYBench is an original and challenging benchmark for measuring the reasoning capabilities of LLMs by leveraging physics problems. As shown in Table 1, PHYBench contains 500 originally curated questions across diverse domains including mechanics, electromagnetism, thermodynamics, optics, modern physics, and advanced physics.

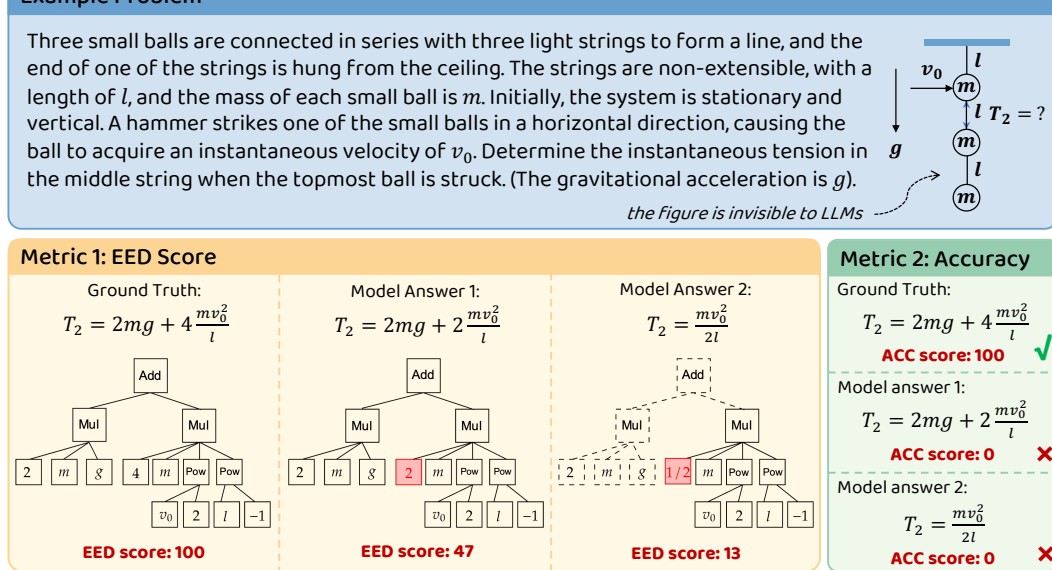

Figure 2: An example problem from PHYBench. Two evaluation metrics are employed: Expression Edit Distance (EED) Score and accuracy. We show the scores for three different responses, with *Model Answer 1* and *Model Answer 2* generated by DeepSeek-R1 and GPT-4o respectively.

An example question is shown in Figure 2. Each question is built around a specific physical scenario, and the model is required to derive a symbolic expression for a key physical quantity based on given conditions. All questions have definitive answers (allowing all equivalent forms, see Section 3.3) and can be solved through physics principles without external knowledge. The challenge lies in the model's ability to construct spatial and interaction relationships from textual descriptions, selectively apply multiple physics laws and theorems, and robustly calculate the evolution and interactions of dynamic systems. Furthermore, most problems involve long-chain reasoning. Models must discard irrelevant physical effects and eliminate non-physical algebraic solutions across multiple steps to prevent an explosion in computational complexity.

Unlike previous reasoning benchmarks that emphasize exhaustive search spaces, PHYBench focuses on realistic physical scenarios that evaluate models' step-by-step physical perception and reasoning abilities. The questions are readily accessible to human experts (with less than 10% of human experts scoring below 30% accuracy), enabling clearer differentiation between models' reasoning capabilities.

## 3.2 Benchmark Curation

All questions in PHYBench are adapted from physics exercises originally designed for human learners, with difficulty levels ranging from high school exercises to Physics Olympiad competitions. To ensure data quality, diversity and validity, we engaged 178 students from Peking University, School of Physics to contribute, adapt, and refine the questions. The overall curation process is illustrated in Figure 3, which consists of two main stages: problem formulation and quality control.

**Problem Formulation.** This stage involves sourcing, adapting, and constructing physics problems suited for evaluation. Our data source includes both non-public and publicly available problems, none of which are easily discoverable through direct internet search or standard references. All problems are text-only without multimodal inputs. During adaptation, each problem is designed as a realistic physical scenario, with a clearly defined target quantity that the solvers must express symbolically using given conditions. For instance, in the mechanics problem shown in Figure 2, the solver is required to analyze the ball's acceleration and derive the expression for the top string's tension: $T = 2mg + 4mv_0^2/l$. To ensure that the correctness of an answer can be determined solely by checking the equivalence of symbolic expressions, the following requirements are enforced during problem construction:

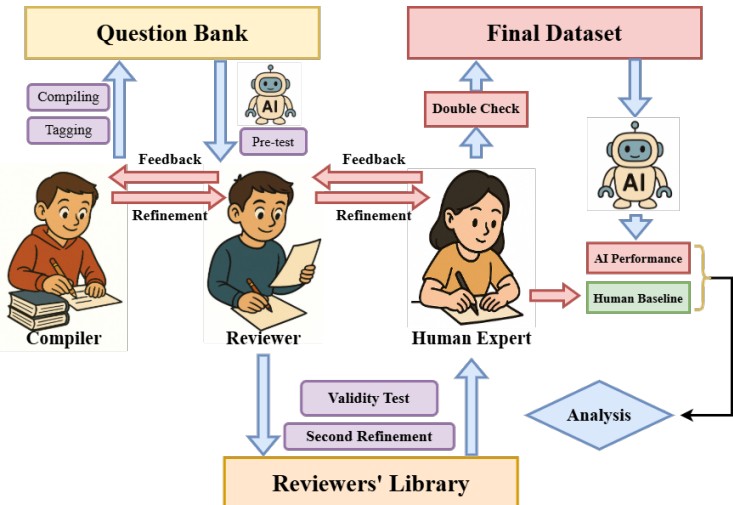

Figure 3: Pipeline of PHYBench data curation.

- **Symbolic-form answer**: Each answer must take the form of a single symbolic expression (e.g., $2mg + 4mv_0^2/l$). We allow all equivalent forms (e.g., factored or rearranged) but reject equations (e.g., $T/m - 2g = v_0^2/l$) or floating-point approximations.
- **Precise statements**: Problem statements must be phrased rigorously to ensure a single unambiguous interpretation and a unique correct solution. All variables must be clearly defined, and the problem should be solvable without requiring any external knowledge or unstated assumptions.

**Quality Control.** Following initial formulation, each question undergoes multiple rounds of review, filtering, and refinement to ensure both data quality and validity. First, all drafted questions are uploaded to an internal *Question Bank* platform. Each question is then assigned to expert reviewers to verify its adherence to construction requirements. If a question fails to meet the standards, reviewers either revise the content directly or return it to the contributor for further editing. To assist this process, we display outputs from several LLMs (including o1 [18] and DeepSeek-R1 [6]) to help reviewers detect ambiguous or misleading statements. All model responses are generated through closed-source APIs under standard zero-shot settings, without access to ground truths or internal annotations. These models are used only for evaluation purposes and are not involved in the construction of the questions. Reviewers iteratively refine the problem statements until the model outputs consistently reflect the intended meaning. Upon approval, the questions are archived in the *Reviewer's Library*.

Finally, we conducted a large-scale human evaluation involving 81 students from Peking University. Among them, 50 participants had achieved gold medal–level performance in the Chinese Physics Olympiad. Each participant independently attempted a subset of the questions and provided feedback on clarity, solution uniqueness, and potential ambiguity. Based on this evaluation, we retained 500 questions from 757 total in *Reviewer's Library*, with a reservation rate of 66.1%. These finalized questions constitute the final PHYBench benchmark. The invited human experts also serve as the human baseline for comparison with model performance, as detailed in Section 4.2.

### 3.3 Evaluation Metric

In this section, we introduce the pipeline and details of the **EED Score**, our automated, model-free metric designed to evaluate the correctness of AI-generated solutions. In Figure 2, we demonstrate how the EED Score assigns partial credit and distinguishes between subtly different outputs. Additional examples and detailed evaluation flow are provided in Appendix B.

The EED Score evaluates the similarity between regularized expression trees derived from model-generated (*gen*) and ground truth (*gt*) expressions. To compute the EED Score, we first convert both *gt* and *gen* expressions from LaTeX into canonical forms using SymPy [14], and then construct their corresponding regularized expression trees. We define the **relative edit distance** $r$ as the number of minimum number of node-level operations (insertions, deletions, or substitutions) required to

transform the *gt* tree into the *gen* tree, normalized by the number of nodes in the *gt* tree. The final EED Score is computed using the extended Zhang-Shasha algorithm [2], defined as follows:

$$
r = \frac{\text{Distance}(T_{\text{gt}}, T_{\text{gen}})}{\text{Size}(T_{\text{gt}})}, \quad \text{score} = \begin{cases} 100, & \text{if } r = 0 \quad (\text{exact match}), \\ 60 - 100r, & 0 < r < 0.6, \\ 0, & r > 0.6. \end{cases} \tag{1}
$$

Function 1 assigns 0 to fully incorrect outputs, while awarding up to 60 points for answers with minor structural or coefficient errors, thereby acknowledging partial correctness. To better capture structural similarity, we extend standard tree-edit operations with **subtree insertions and deletions**, assigning a cost equivalent to 60% of the standard operation cost for subtrees with more than five nodes. This allows the algorithm to more efficiently align structurally similar though not identical expressions.

Furthermore, in Appendix B, we present two key insights on the EED Score. First, we demonstrate that EED Score significantly improves sample efficiency: our 500-problem benchmark, when scored with EED, achieves discriminative power comparable to that of 1500 problems evaluated with traditional accuracy-based scoring. Second, we conduct a robustness analysis by varying the baseline score (default: 60) and the penalty coefficient (default: 100) in the scoring function. This analysis shows that EED Score remains stable and reliable across a range of parameter settings.

## 4 Experiments

In this section, we evaluate a set of LLMs on the PHYBench benchmark, covering both state-of-the-art models and widely used baselines. A human baseline is also included for comparison. Our evaluation aims to determine: (1) Whether current reasoning models can match or exceed human expert performance; (2) Whether PHYBench can reliably distinguish between models' reasoning capabilities; (3) Whether our dataset is robust under TTS conditions.

### 4.1 Experiment Setup

**Baseline Models.** We evaluate a diverse set of models, including state-of-the-art models as well as other widely adopted or representative models. For API-based evaluations, we include GPT-4o [16], GPT-4.1 [19], o1 [17], o3-mini [21], o3 [20], o4-mini [20], Claude 3.7 Sonnet [3], Claude 3.7 Sonnet Thinking [3], Gemini 2.0 Flash Thinking [25], Gemini 2.5 pro [25], DeepSeek-V3 [5], DeepSeek-R1 [6], Qwen2.5-max [26], Grok 3 Beta [9]. The remaining models (DeepSeek-R1-Distill-Qwen-32B [6] and QwQ-32B [27]) are evaluated locally.

**Evaluation Details.** We employ both **accuracy** and **EED Score**, as detailed in Section 3.3. API evaluations use the default hyperparameters of each service. For locally evaluated models, we set `temperature` to 0.6, `top_p` to 0.95, and `max_tokens` to 32,768. The detailed prompts are shown in Appendix D. We use four NVIDIA A100 Tensor Core GPUs with 80GB memory for inference.

### 4.2 Human Baseline

We recruited 81 students from Peking University, School of Physics. Among them, 50 participants were gold medalists in the Chinese Physics Olympiad. Every student is assigned eight problems from the PHYBench dataset. In total, we obtained 559 valid answer sheets corresponding to problems within the scope of the publicly released PHYBench dataset. Human performance averaged an accuracy of $61.9 \pm 2.1\%$ and an EED Score of $70.4 \pm 1.8$, where the uncertainties were estimated from 10,000 bootstrap resamples. At the 99% confidence level, experts significantly outperformed all evaluated LLMs on both metrics. Moreover, the upper quartile of the human score distributions reached 71.4% for accuracy and 80.4 for the EED Score.

### 4.3 Main Results

We assessed several models on the PHYBench dataset, using both **accuracy** and the **EED Score** as evaluation metrics. Their performances are summarized in Figure 1.

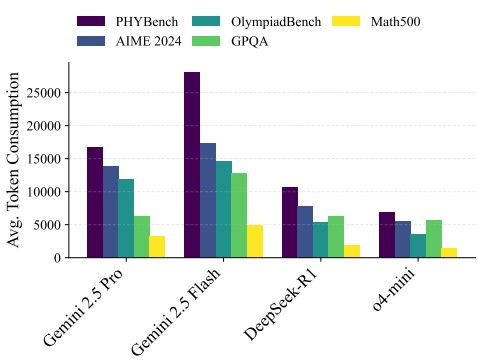

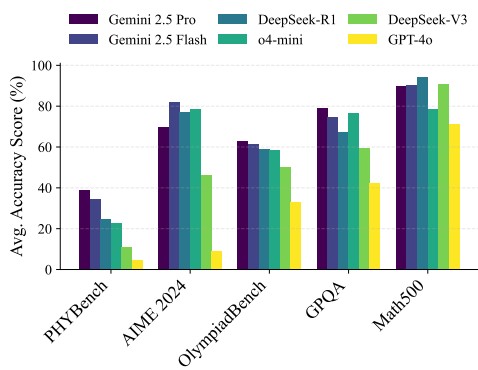

(a) Model Token Usage Across Benchmarks

(b) Score of Models on Different Benchmarks.

Figure 4: Token Usage and Score of Typical Models on Different Benchmarks

The highest-performing model, Gemini 2.5 Pro, attains an accuracy of 36.9% and an EED Score of 49.5, which remains significantly below the human baseline. Notably, reasoning models generally outperform base models. Recent general-purpose models, such as DeepSeek-V3 [5], Claude 3.7 Sonnet [3] and GPT-4.1 [19], achieve relatively strong results with accuracies of 13.6%, 13.2% and 12.9% respectively. In contrast, 32B models including DeepSeek-Distill-32B and QwQ-32B demonstrate substantially weaker performance, with accuracies of 2.6% and 1.2% and EED Scores of 4.5 and 3.2 respectively—despite their strong performances on other benchmarks [6, 27]. Their limited performance on PHYBench may be attributed to either the long-horizon nature of PHYBench tasks or the physical perception challenge beyond conventional QA settings.

While accuracy and the EED Score yield nearly identical model rankings, our analysis reveals the EED Score as a superior evaluation metric due to its broader score distribution and lower statistical uncertainty. Our bootstrap analysis (see Appendix C) reveals that EED Score improves sample efficiency by an average of 204% with a standard deviation of 80%. In other words, evaluating on 500 problems with EED Score provides discriminatory power equivalent to approximately 1500 problems with binary accuracy scoring. This improvement allows for a more consistent and reliable evaluation.

## 4.4 Comparison with Other Benchmarks

To quantify the difficulty and characteristics of PHYBench, we compare it with several widely-used reasoning benchmarks, including MATH-500 [13], AIME 2024 [11], OlympiadBench [10], and GPQA [24]. The details of the experimental setup are provided in Appendix E.

As shown in Figure 4, PHYBench requires significantly more output tokens on average compared to other benchmarks, indicating longer and more complex reasoning chains. At the same time, model scores on PHYBench are consistently lower than on other benchmarks, especially for non-reasoning models. These results reflect the higher complexity and difficulty of PHYBench.

In addition, PHYBench shows clearer performance separation between reasoning and non-reasoning models. The gap between reasoning models like DeepSeek-R1 and general models like DeepSeek-V3 is much larger on PHYBench than on other datasets. This makes PHYBench more effective at distinguishing reasoning capacity. As discussed in Appendix A, our dataset avoids many of the noise issues commonly found in other benchmarks, leading to more reliable score comparisons.

## 4.5 Test Time Scaling on PHYBench

We further examined TTS behavior of models on PHYBench, with detailed methodology provided in Appendix E. As shown in Figure 5a, the pass@$k$ accuracy improves smoothly as $k$ increases, while maintaining **order-preservation**: models with better single-sample performance continue to outperform others under scaling. Figure 5b further confirms that the separation between model capabilities remains pronounced through majority voting scaling. The extrapolated upper bounds for

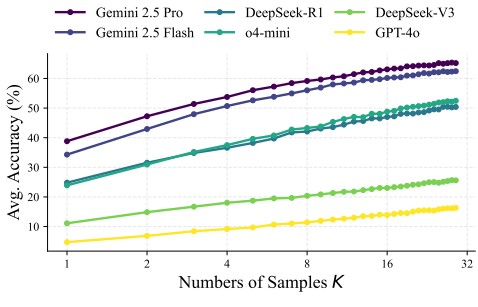
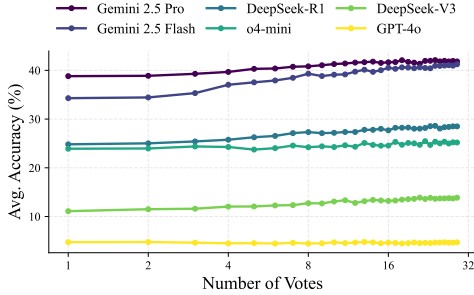

(a) pass@$k$ accuracy on PHYBench.

(b) Majority voting accuracy on PHYBench.

Figure 5: TTS on PHYBench: comparison between pass@$k$ and majority voting strategies, both evaluated under varying numbers of sampled responses $k$ (log-scale on the x-axis).

each model are provided in Table 7. It is shown that Gemini 2.5 Flash closes the gap with Gemini 2.5 Pro, while DeepSeek-R1 continues to outperform o4-mini more clearly.

## 5 Error Analysis

PHYBench problems are multi-condition and multi-step in nature, requiring models to construct long and complex reasoning chains. Leveraging this characteristic, we conduct two complementary analyses that clarify **where** and **why** modern language models fail: **(1) Stage-wise error localization** decomposes the reasoning process into distinct steps and dimensions, allowing us to pinpoint which stage contributes most to model failure. **(2) Proof of superficial reasoning** defines and empirically confirms that models often rely on pattern matching rather than genuine understanding.

### 5.1 Stage-wise Failure Localization

**Step 1: Physical Perception (PP) versus Robust Reasoning (RR).** We locate the first mistake of each reasoning trace by seven models across 50 representative problems. If the error stems from a failure to abstract the physical scenario—such as misidentifying key variables, overlooking relevant quantities, or misunderstanding their relationships—we categorize it as a PP error. Other errors are classified as RR, which include selecting inappropriate formulas, or failing to combine given conditions to complete the derivation. Figure 12 illustrates typical examples of both error types. As shown in Table 2, typically more than 90% of the observed errors occurred during RR, indicating that most failures arise after the physical scenario has already been correctly understood.

**Step 2: Semantic versus Symbolic Reasoning.** To further analyze RR errors, we divide them into two categories. **Semantic reasoning** involves generating new equations not directly entailed by previous ones, typically by interpreting the problem statement or applying physical laws. In contrast, **symbolic reasoning** refers to manipulating existing equations to derive logical consequences, such as simplification or substitution. As shown in Table 2, over 90% of RR errors fall into the semantic category, suggesting that models struggle primarily with non-formulaic aspects during reasoning.

These two axes of analysis localize the majority of model errors to the domain of **semantic reasoning**. This suggests that models are generally reliable in interpreting given physical conditions and performing symbolic manipulations between established equations, but often struggle when deriving new, non-entailed equations from the physical context and problem description. For example, models may incorrectly assume angular momentum conservation even when external torques from magnetic fields are present. This indicate that current models fail to grasp the underlying physical principles.

### 5.2 Superficial Reasoning and Robustness of Reasoning

We define **superficial reasoning** as reasoning processes driven by pattern matching in the context. It manifests as the model retrieving a known mapping to the answer without grasping the physical context. While superficial reasoning allows models to perform complex and precise symbolic derivations, it lacks robustness when faced with unfamiliar or perturbed inputs.

Table 2: Error distribution statistics for all models. **PP** and **RR** represent the proportion of two error types at the first mistake; **Sem** and **Sym** denote, among RR errors, the proportion of **semantic** and **symbolic** reasoning errors, respectively. All values are percentages.

| Metric (%) | Gemini 2.5 Pro | DeepSeek-R1 | DeepSeek-V3 | o4 mini | o3 mini | o1-preview | GPT-4o |
|---|---|---|---|---|---|---|---|
| Accuracy | 40 | 27 | 14 | 27 | 19 | 18 | 5 |
| PP | 9 | 4 | 5 | 6 | 10 | 12 | 21 |
| RR | 91 | 96 | 95 | 94 | 90 | 88 | 79 |
| Sem | 94 | 91 | 87 | 99 | 99 | 95 | 90 |
| Sym | 6 | 9 | 13 | 1 | 1 | 5 | 10 |

To expose superficial reasoning, we conduct a perturbation experiment. We provide each model with a partial solution trace and inject a deliberate error into each (see Appendix G for details). Each model is required to continue the derivation. We assess reasoning robustness by examining whether the model can detect and correct the injected error; blindly continuing the flawed reasoning serves as a clear signal of superficial reasoning.

By analyzing how models continue from a perturbed reasoning trace, we identify three distinct reasoning modes: **superficial reasoning**, **genuine reasoning**, and **pseudo-genuine reasoning**, all of which are illustrated in detail in Appendix G.3.

**Superficial reasoning** blindly continues the flawed trace without verification, failing to detect or correct the injected error. This mode is highly vulnerable to all perturbations.

**Genuine reasoning** identifies the flaw and repairs it through semantic understanding—e.g. correcting $R - h$ to $R + h$ after recognising the geometric definition of altitude. This mode exhibits strong robustness across all types of perturbations.

**Pseudo-genuine reasoning** detects and corrects some errors through automatic consistency checks, such as dimensional analysis or limiting-case evaluation. While this approach offers partial robustness, it does not consistently handle all types of perturbations.

Table 3: Accuracy (%) of models under different settings. Original: solving without trace; Correct: given a correct partial trace. T1–T6: different perturbation types (see Appendix G.2).

| Model | Original | Correct | T1: dim | T2: $\pm$ | T3: 1+2 | T4: miss $h$ | T5: 2+4 | T6: formula |
|---|---|---|---|---|---|---|---|---|
| Gemini 2.5 Pro | 97 | 100 | 93 | 95 | 100 | 78 | 95 | 100 |
| DeepSeek-R1 | 97 | 98 | 64 | 39 | 99 | 37 | 78 | 94 |
| DeepSeek-V3 | 66 | 93 | 0 | 97 | 73 | 0 | 0 | 12 |
| o3 mini | 98 | 98 | 88 | 85 | 97 | 73 | 90 | 95 |
| o4 mini | 83 | 89 | 55 | 70 | 72 | 34 | 54 | 90 |
| o1-preview | 94 | 81 | 9 | 15 | 70 | 10 | 14 | 83 |
| GPT-4o | 4 | 0 | 0 | 0 | 0 | 0 | 0 | 1 |

Table 3 summarises performance drops under six perturbation types. Non-reasoning models are highly vulnerable across all perturbations. Early reasoning models like o1-preview also shows less robustness. In contrast, recent reasoning models such as DeepSeek-R1 and Gemini 2.5 Pro exhibit significantly greater robustness—but largely through **compensatory strategies** rather than genuine semantic understanding. DeepSeek-R1 relies on symbolic checks such as dimensional analysis and limiting-case evaluation to detect flaws. While effective against symbolic perturbations, it becomes vulnerable when such cues are absent, as in T2 and T4. Gemini 2.5 Pro avoids semantic reasoning by shifting to formal derivations, thus reducing reliance on physical interpretation and maintaining perturbation robustness within 8 percentage points. Such pseudo-genuine fixes increase resilience without addressing the core semantic bottleneck.

**Implications for future work.** The gap between superficial robustness and true semantic competence remains wide. With long-horizon problems and targeted perturbation protocol, PHYBench offers a principled testbed for guiding models toward genuine physical understanding.

# 6   Conclusion and Limitations

This paper introduces PHYBench, an original and challenging benchmark with 500 carefully curated physics problems for evaluating the reasoning capabilities of LLMs. We also propose the EED Score, a fine-grained metric for evaluating symbolic expressions. Evaluations demonstrate that

PHYBench is challenging, robust under TTS and effectively differentiates models. The results show that even state-of-the-art models fall far behind human experts on PHYBench. Moreover, current LLMs struggle with multi-step and multi-condition inference, introducing incorrect equations and lacking the ability to identify or correct such errors.

Regarding limitations, our problems' primary focus on Olympiad-level difficulty and uneven distribution across diverse physics topics limit generalization to research-level reasoning. Additionally, the EED Score focuses on final answer quality and does not capture the full reasoning process. Future work will expand the dataset in both scale and coverage, with greater emphasis on evaluating intermediate steps to enable more consistent and detailed assessment.

# 7  Contributions and Acknowledgements

PHYBench was constructed with strong support from the School of Physics at Peking University, Ministry of Education Physics 101 Plan, and National Science Foundation of China under contract No. 12425505, 12235001, U2230402. In total, more than a hundred students in the School have participated in this project and made valuable contributions. The PHYBench project aspires to lead the development of LLM by using high-quality physics benchmarks and data-driven to reveal the nature of AI's understanding and reasoning in the physical world and in the face of complex problems.

Our team members contribute to the development of PHYBench from the following perspectives:

- Research Pipeline Construction
- Data Annotation
- Data Quality Inspection

- Model Evaluation
- Result Analysis
- Paper Writing

**Core Contributors**

- **Shi Qiu**
- **Shaoyang Guo**
- **Zhuo-Yang Song**
- **Yunbo Sun**
- **Zeyu Cai**
- **Jiashen Wei**
- **Tianyu Luo**

- **Yixuan Yin**
- **Haoxu Zhang**
- **Yi Hu**
- **Chenyang Wang**
- **Chencheng Tang**
- **Haoling Chang**
- **Qi Liu**

- **Ziheng Zhou**
- **Tianyu Zhang**
- **Jingtian Zhang**
- **Zhangyi Liu**
- **Minghao Li**
- **Yuku Zhang**
- **Boxuan Jing**

**Contributors**

- **Xianqi Yin**
- **Yutong Ren**
- **Zizhuo Fu**
- **Jiaming Ji**
- **Weike Wang**
- **Xudong Tian**
- **Laifu Man**
- **Jianxiang Li**
- **Feiyu Tao**
- **Xiaotian Li**
- **Xianqi Xia**
- **Jiawei Lin**
- **Zheyu Shen**

- **Jiahang Chen**
- **Qiuhao Xiong**
- **Binran Wang**
- **Fengyuan Wang**
- **Ziyang Ni**
- **Bohan Zhang**
- **Fan Cui**
- **Changkun Shao**
- **Bozu Zhang**
- **Lixiang Tang**
- **Zekai Zhao**
- **Heyun Zou**
- **Zan Lou**

- **Yizhe Tian**
- **Chenxu Yu**
- **Wenshuai Liu**
- **Yantong Wang**
- **Dihang Sun**
- **Hanyu Cao**
- **Yuchen Lu**
- **Haoyu Mo**
- **Shuran Yang**
- **Qianyi Wang**
- **Zhiyuan Zhou**
- **Yuxin He**
- **Anqi Lv**

- **Yifan Shi**
- **Zijian Wang**
- **Jinyu Zhou**
- **Zhiji Feng**
- **Xinlin Zhu**
- **Yixin Liu**
- **Zihan Tang**
- **Boqian Yao**
- **Jiawei Chen**
- **Tianxing Huang**
- **Boxun Yu**
- **Zihao Xu**
- **Rundong Liu**
- **Xuqi Jiang**
- **Haoxiang Li**
- **Wei Yan**
- **Aoqin Liang**
- **Zirui Peng**
- **Tianxiao Li**
- **Jiarui Tang**
- **Yuyang Weng**
- **Chen Huang**
- **Yiwei Deng**
- **Qihang Li**
- **Yuntian Xie**
- **Chengkai Sheng**
- **Xianhong Zeng**
- **Yizhe Zheng**
- **Bowen Yu**
- **Chengzhou Wu**
- **Mengyao Zhang**
- **Houcheng Li**
- **Peilin Li**
- **Yuyang Zhao**
- **Bingru He**
- **Zongyue Hou**
- **Jiajun Yan**
- **Lingrui Zhang**
- **Jianyuan Luo**

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

# List of appendices

# Appendices

## A  Detailed Analysis of Limitations in Existing Reasoning Benchmarks

In this section, we provide an extended discussion of the three key limitations identified in Section 1 that hinder the effectiveness of current reasoning benchmarks. We present detailed examples along with statistical evidence illustrating each limitation. These cases highlight the need for PHYBench, which is designed to address these issues through original and challenging physics problems with careful calibration. The examples are annotated to highlight observed errors and deficiencies.

### A.1  Oversimplified Reasoning Tasks

State-of-the-art reasoning models exhibit performance saturation on traditional benchmarks. When scores are already high, the differences between models become small and less meaningful. During our experiments, we observed that certain benchmarks, such as MATH-500 [13], are sensitive to minor formatting issues—for example, whether models include units in their answers. These are not failures in reasoning, but issues with instruction adherence. After simple answer-format corrections, models like Gemini 2.5 Pro [25], o4 mini-high [20] and DeepSeek-R1 [6] produce entirely correct answers, suggesting that such benchmarks may no longer effectively differentiate reasoning capabilities.

To further investigate this issue, we examined existing datasets, using GPQA [24] as a representative example. We selected two physics questions directly from the original paper, detailed as follow. Our analysis shows that, despite their uncommon topic coverage, these questions mainly test factual knowledge rather than requiring long or complex reasoning chains. This helps explain the generally low reasoning-token counts observed among many reasoning benchmarks, as shown in Table 1.

---

**GPQA Selected Problem–Astrophysics**

Astronomers are studying a star with a $T_{\text{eff}}$ of approximately 6000 K. They are interested in spectroscopically determining the surface gravity of the star using spectral lines (EW$< 100$ mÅ) of two chemical elements, El1 and El2. Given the atmospheric temperature of the star, El1 is mostly in the neutral phase, while El2 is mostly ionized. Which lines are the most sensitive to surface gravity for the astronomers to consider?

(A) El2 I (neutral)
(B) El1 II (singly ionized)
(C) El2 II (singly ionized)
(D) El1 I (neutral)

**Solution.**  The sensitivity to $\log g$ comes from the pressure dependence of the ionization balance (via the Saha equation)

$$\frac{n_{\text{II}}}{n_{\text{I}}} \;\propto\; \frac{T^{3/2}}{P_e} \exp\!\left(-\tfrac{\chi}{kT}\right),$$

so the minority species population (where $n_{\text{II}} \ll n_{\text{I}}$ or vice versa) changes most with electron pressure $P_e$. Since El1 is mostly neutral, its El1 II lines are the minority species and thus most gravity-sensitive.

$\boxed{\text{(B) El1 II}}$

---

**GPQA Selected Problem–Quantum Mechanics**

Suppose we have a depolarizing channel operation given by $E(\rho)$. The probability $p$ of depolarization represents the strength of the noise. If the Kraus operators of the channel are

$$A_0 = \sqrt{1 - \frac{3p}{4}}, \quad A_1 = \sqrt{\frac{p}{4}}\, X, \quad A_2 = \sqrt{\frac{p}{4}}\, Y, \quad A_3 = \sqrt{\frac{p}{4}}\, Z,$$

---

> what could be the correct Kraus representation of the map $E(\rho)$?
>
> $$\text{(A) } E(\rho) = (1-p)\,\rho \;+\; \frac{p}{3}\,X\rho X \;+\; \frac{p}{3}\,Y\rho Y \;+\; \frac{p}{3}\,Z\rho Z,$$
>
> $$\text{(B) } E(\rho) = (1-p)\,\rho \;+\; \frac{p}{3}\,X\rho^2 X \;+\; \frac{p}{3}\,Y\rho^2 Y \;+\; \frac{p}{3}\,Z\rho^2 Z,$$
>
> $$\text{(C) } E(\rho) = (1-p)\,\rho \;+\; \frac{p}{4}\,X\rho X \;+\; \frac{p}{4}\,Y\rho Y \;+\; \frac{p}{4}\,Z\rho Z,$$
>
> $$\text{(D) } E(\rho) = (1-p)\,\rho^2 \;+\; \frac{p}{3}\,X\rho^2 X \;+\; \frac{p}{3}\,Y\rho^2 Y \;+\; \frac{p}{3}\,Z\rho^2 Z.$$
>
> **Solution.** By definition
>
> $$E(\rho) = \sum_{i=0}^{3} A_i\,\rho\,A_i^\dagger = \left(1 - \tfrac{3p}{4}\right)\rho + \frac{p}{4}\left(X\rho X + Y\rho Y + Z\rho Z\right).$$
>
> Re-parameterizing the "depolarization probability" so that $p_{\text{eff}} = 3p/4$ yields the standard form
>
> $$E(\rho) = (1 - p_{\text{eff}})\,\rho + \frac{p_{\text{eff}}}{3}\left(X\rho X + Y\rho Y + Z\rho Z\right),$$
>
> which matches choice (A).
>
> $$\boxed{\text{(A)}}$$

## A.2 Potential Data Contamination

Many existing benchmarks are built from publicly available sources, including web pages, e-books, and released exam questions. Such content may have already been included **in the pretraining data of large language models**, leading to potential data leakage.

We consider AIME 2024 [11] a high-quality and challenging benchmark. As shown in Table 1, the average output length of models on AIME 2024 is second only to PHYBench, and significantly higher than on other reasoning benchmarks. This suggests that solving these problems requires extended reasoning and detailed step-by-step explanation.

However, in our evaluation, Gemini 2.5 Flash achieved **100% accuracy** on AIME 2024, with an average score **above 99%** across 16 independent runs. This raises concerns that the model may have memorized parts of the dataset, rather than truly mastering generalizable reasoning strategies. Furthermore, in Section 5, our reasoning robustness experiments further show that chat-based models are highly sensitive to small perturbations in the reasoning process, suggesting a lack of robustness and deeper conceptual understanding.

## A.3 Lack of Rigorous Verification

Existing reasoning benchmarks often lack sufficient verification and validation procedures. For high-quality problems that are both original and complex, ensuring the correctness, solvability, and clarity of the questions becomes significantly more difficult. This raises the bar for human-level validation. Even for problems adapted from public sources, multiple rounds of review are necessary to eliminate instruction-following ambiguities and format-related inconsistencies.

In our dataset comparison experiment (Section 4.4), we observed concrete verification issues in OlympiadBench. Specifically, we closely examined two physics problems and identified critical flaws. **Problem 1015** includes a physical quantity $\gamma$ in the answer that was never mentioned in the problem statement. In **Problem 1216**, the ground truth is incorrectly extracted, causing all model outputs, while mostly correct during experiment, to be falsely judged.

To better quantify such issues, we conducted a statistical analysis. As described in Appendix E, we randomly sampled 36 physics problems from OlympiadBench where the reference answers are symbolic expressions. Among these, 14 problems exhibited questionable answer quality—either due to ambiguous phrasing or errors in answer extraction. These findings underscore the challenges of properly calibrating high-difficulty benchmarks and highlight the importance of rigorous data validation, especially when evaluating models on complex reasoning tasks.

## Problem 1015–Missing $\gamma$ variable

**Question (2.4).** Find the minimum velocity $u$ of an updraught (air flowing upwards) that will keep the bubble from falling at thermal equilibrium. Give your answer in terms of $\rho_s, R_0, g, t$ and the air's coefficient of viscosity $\eta$. You may assume that the velocity is small such that Stokes's law applies, and ignore the change in the radius when the temperature lowers to the equilibrium. The drag force from Stokes' Law is

$$F = 6\pi \eta R_0 u.$$

**Context.** An Electrified Soap Bubble

- A spherical soap bubble with internal air density $\rho_i$, temperature $T_i$ and radius $R_0$ is surrounded by air with density $\rho_a$, atmospheric pressure $P_a$ and temperature $T_a$. The soap film has surface tension $\gamma$, density $\rho_s$ and thickness $t$. Assume $R_0 \gg t$.
- The increase in energy $dE$ needed to increase the surface area of a soap–air interface by $dA$ is given by
$$dE = \gamma \, dA.$$

**Earlier context questions:**

1. Find $\dfrac{\rho_i T_i}{\rho_a T_a}$ in terms of $\gamma$, $P_a$ and $R_0$.

2. Compute the numerical value of $\dfrac{\rho_i T_i}{\rho_a T_a} - 1$ using $\gamma = 0.0250 \, \mathrm{N\,m^{-1}}$, $R_0 = 1.00 \, \mathrm{cm}$, $P_a = 1.013 \times 10^5 \, \mathrm{N\,m^{-2}}$.

3. If the bubble is initially formed with warmer air inside, find the minimum numerical value of $T_i$ so that the bubble can float in still air. Use $T_a = 300 \, \mathrm{K}$, $\rho_s = 1000 \, \mathrm{kg\,m^{-3}}$, $\rho_a = 1.30 \, \mathrm{kg\,m^{-3}}$, $t = 100 \, \mathrm{nm}$, and $g = 9.80 \, \mathrm{m\,s^{-2}}$.

4. After thermal equilibration, the bubble in still air will naturally fall toward the ground.

**Answer:**
Ignore the radius change $\rightarrow$ radius remains $R_0$.
The drag force from Stokes' Law is
$$6\pi \eta R_0 u.$$
At equilibrium, the upward drag balances the net weight minus buoyant force,

$$6\pi \eta R_0 u \geq \left(4\pi R_0^2 \rho_s t + \tfrac{4}{3}\pi R_0^3 \rho_i\right)g - \tfrac{4}{3}\pi R_0^3 \rho_a g.$$

Since in thermal equilibrium $T_i = T_a$ and $\rho_i = \rho_a\left(1 + \frac{4\gamma}{R_0 P_a}\right)$, we have

$$6\pi \eta R_0 u \geq \left(4\pi R_0^2 \rho_s t + \tfrac{4}{3}\pi R_0^3 \rho_a\left[1 + \frac{4\gamma}{R_0 P_a}\right]\right)g - \tfrac{4}{3}\pi R_0^3 \rho_a g.$$

Rearranging gives the minimum updraught speed

$$\boxed{u \geq \frac{4R_0 \rho_s t g}{6\eta} + \frac{\tfrac{4}{3}R_0^2 \rho_a g\left(\frac{4\gamma}{R_0 P_a}\right)}{6\eta}.}$$

**Model Answers (Actually correct)**

$$\boxed{u = \frac{2\rho_s R_0 g t}{3\eta}} \qquad \textbf{Equal as} \qquad \boxed{u = \frac{2R_0 t \rho_s g}{3\eta}}$$

## Problem 1216–Wrongly extracted answer

**Context (excerpt).** An accelerated charged particle radiates electromagnetic energy. The radiated power $P_{\mathrm{rad}}$ of a charged particle that moves on a circular path with constant angular velocity is assumed to depend only on

$a$   (centripetal acceleration),      $q$   (particle charge),

$c$   (speed of light),             $\varepsilon_0$   (vacuum permittivity).

**Question (A.4).** Use dimensional analysis to find an expression for the radiated power $P_{\mathrm{rad}}$.

**Solution (outline).** Assume a power-law form

$$P_{\mathrm{rad}} \;=\; a^{\alpha}\, q^{\beta}\, c^{\gamma}\, \varepsilon_0^{\delta},$$

and equate the SI base-unit dimensions on both sides to determine the exponents $\alpha, \beta, \gamma, \delta$.

**Final answer (Wrongly extracted)**

$$\boxed{P_{\mathrm{rad}} \;=\; a^{\alpha}\, q^{\beta}\, c^{\gamma}\, \varepsilon_0^{\delta}}$$

(with specific values of $\alpha, \beta, \gamma, \delta$ fixed by dimensional consistency).

**Model Answers (Actually correct)**

$$\boxed{P_{\mathrm{rad}} = \frac{K\, q^2\, a^2}{\varepsilon_0\, c^3}} \qquad \textbf{Equal as} \qquad \boxed{P_{\mathrm{rad}} = C\, \frac{q^2\, a^2}{\varepsilon_0\, c^3} \;\approx\; \frac{q^2\, a^2}{6\pi\, \varepsilon_0\, c^3}}$$

# B   Evaluation Metric

## B.1   Tree Editing Distance Algorithm

This section demonstrates details and principles of our EED scoring metric's operational pipeline. The pipeline initiates by extracting the final \boxed{} component from the input string-formatted LaTeX expression. Subsequently, a series of preprocessing procedures (e.g., removing formatting commands and complete begin...end environments) are applied, normalizing non-standard LaTeX expressions to a parser-compatible form.

Next, we utilize a Python library called latex2sympy_extended [12] to translate the normalized LaTeX into a symbolic expression compatible with SymPy [14]. For computational efficiency during simplification, we assume all symbolic variables to be positive. The simplify() function is then applied individually to both the *gt* and *gen* expressions.

A solution is considered fully correct if the simplified *gt* and *gen* expressions are equivalent, which is checked through the **equals** method, determining the equivalence of expressions by combining symbolic simplification and numerical verification. For **accuracy** metric, our evaluation formula is simply defined as follows:

$$\text{score}_{\mathrm{ACC}} = \begin{cases} 100, & \text{if } \texttt{equals}(\texttt{simplify}(\text{gt}), \texttt{simplify}(\text{gen})) = \texttt{True}, \\ 0, & \text{otherwise.} \end{cases} \tag{2}$$

However, unlike conventional benchmarks that employ binary scoring based on final results, our EED scoring proposes a model-free partial credit mechanism to better reflect solution correctness in symbolic mathematics. For detailed illustration, consider an electromagnetic problem where *gt* is:

$$B = \sqrt{\frac{n_2^2}{n_1^2} + \frac{1}{2}} \frac{4mQ}{\pi \epsilon_0 a^3 q} \tag{3}$$

Two incorrect generated answers may demonstrate fundamentally different understanding levels:

- **Coefficient error**: $B = \sqrt{\frac{n_2^2}{n_1^2} + \frac{1}{2}} \frac{2mQ}{\pi \epsilon_0 a^3 q}$
- **Structural error**: $B = \frac{\pi Q q}{n_1 n_2 a}$

The former preserves the solution's physical essence with minor computational errors, while the latter indicates a fundamental misunderstanding. To quantify this distinction, we implement an **extended tree editing distance metric** for similarity assessment, with a detailed illustration in Figure 6.

In `SymPy`'s expression tree representation, fundamental mathematical components (constants, variables, operators, functions) constitute a tree structure. Following the conversion of `SymPy` expressions into trees, we calculate the minimum editing distance between gt and gen trees through a sequence of basic node operations (insertions, deletions, and updates) with specific cost. This edit distance metric effectively quantifies structural dissimilarity between expressions. The implementation leverages the dynamic programming-based Zhang-Shasha algorithm [2], which exhibits a time complexity of $O(n_1 n_2 d_1 d_2)$ and space complexity of $O(n_1 n_2)$ where $n_{12}, d_{12}$ denote the node count and maximum depth of respective trees. For our specific expression tree editing problem, these computational requirements remain entirely acceptable compared to the time cost of `simplify()` method.

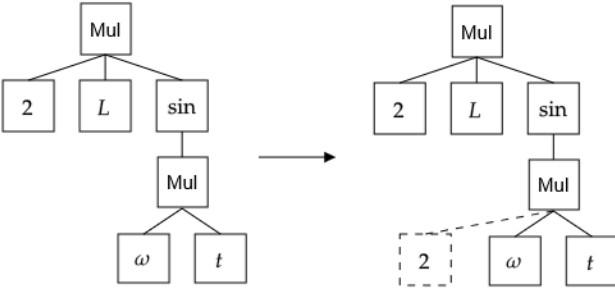

Figure 6: An example of expression tree editing from $2L \sin \omega t$ to $2L \sin 2\omega t$. Numbers, variables, functions and fundamental binary operations are regarded as tree nodes.

The score is then determined by the **relative editing distance**, $r$, which is the ratio of the editing distance to the tree size. If any error occurs during formatting, conversion, or computation procedures, the returned score will be set to zero due to the model's incorrect input format, a phenomenon particularly prevalent among distilled models. We restate our scoring function as follows:

$$r = \frac{\text{Distance}(T_{\text{gt}}, T_{\text{gen}})}{\text{Size}(T_{\text{gt}})}, \quad \text{score} = \begin{cases} 100, & \text{if } r = 0 \quad \text{(exact match)}, \\ 60 - 100r, & 0 < r < 0.6, \\ 0, & r > 0.6. \end{cases} \quad (4)$$

Additionally, in realistic physics scenarios, a final expression can be factorized into a sum or product of several terms or factors with different physical meanings. For instance, a standard formulation for electric potential typically comprises three principal components: an external field term, a charge distribution term, and an electric dipole moment term, each representing distinct physical contributions to the overall potential field, with an example as follows:

$$V(r) = -E_0 r \cos \theta + \frac{Q}{4\pi\epsilon_0 r} + \frac{p \cos \theta}{8\pi\epsilon r^2} \quad (5)$$

We then introduce a **cluster editing discount** to quantify the correctness of physical components. If a $gen$ expression ignores some components but contains other components correctly, its score is expected to be higher for its correct calculation on some discrete parts of the overall contribution. Consequently, the "clustered mistakes", which often relate to a whole component, should have a discount on their total insertion or deletion cost. For this reason, our tree editing algorithm is extended with two additional operations: **inserting and removing a subtree**, which is illustrated in Figure 7.

We set the cost function of inserting or removing a subtree $T$ with size $x$ to be:

$$\text{Cost}(\texttt{InsertTree}(T), \texttt{DeleteTree}(T)) = \min(x, 0.6(x - 5) + 5) \quad (6)$$

The formula degenerates back to the original cost for $x \leq 5$, reducing the computational expense of term deletion and insertion operations while ensuring the corresponding score remains zero when the entire formula is either deleted or inserted. Notably, this mechanism can also be implemented through extended Zhang-Shasha algorithm [2], preserving identical time and space complexity characteristics.

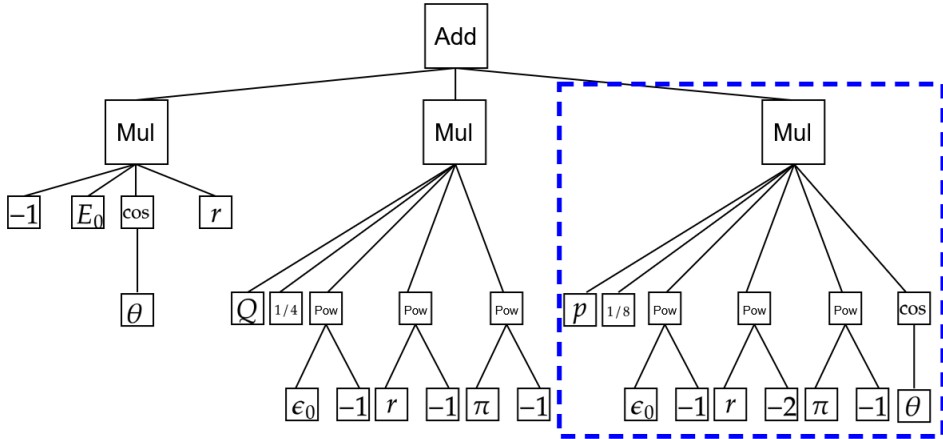

Figure 7: An Example of removing a subtree cluster (subtree in red box) corresponding to an electric dipole moment contribution. We introduce a cluster editing discount to reduce the cost of such an operation since it corresponds to whole physical components.

## B.2 Qualitative Interpretations for Advantages of the EED Score

Traditional binary scoring, which considers only final correctness, fails to effectively capture model performance when tasks are overly easy or difficult. In such cases, scores tend to cluster near the extremes, reducing discriminative power and increasing statistical uncertainty. In contrast, our EED Score provides a finer-grained evaluation that mitigates this issue by offering more informative and continuous measurements of solution quality.

To illustrate that the EED Score offers a more discriminative and nuanced evaluation, we construct a simple theoretical model. Considering quantifying the model's physical ability and problem difficulty using real-valued parameters $a$ and $d$ respectively. The corresponding score $s = f(a - d)$ is then determined by a function of their difference.

Under binary scoring, the system operates under an all-or-nothing principle: the model receives full credit only when its ability strictly exceeds the problem's difficulty threshold (i.e., $a > d$). Otherwise, it scores zero. This scoring function can be represented using the Heaviside step function:

$$f_{\text{BIN}}(x) = \theta(x) = \begin{cases} 1 & \text{if } x \geq 0 \\ 0 & \text{otherwise} \end{cases} \tag{7}$$

For our EED scoring, even if the model answer is incorrect, a partially correct answer can still get a non-zero score, which can be approximately described as a linear function.

$$f_{\text{EED}}(x) = \begin{cases} 1, & \text{if } x \geq 0, \\ \max\left(0, 0.6 + 0.01x\right), & \text{otherwise.} \end{cases} \tag{8}$$

In typical benchmarks, problem difficulty can be modeled by a Gaussian distribution with given mean and variance. A higher mean corresponds to greater overall difficulty, while a larger variance indicates more diverse problem difficulty. The relationship between the model score and its ability can be expressed as the convolution of the scoring function and the difficulty distribution function within a fundamental calculation. Furthermore, a benchmark's capacity to differentiate model abilities, referred to as "discrimination", can be characterized by the derivative of the score-ability function. The numerical results are presented below.

$$S(a) = f_{\text{score}} \otimes N_{\text{diff}}(\mu, \sigma^2), \ \text{Dis} = \frac{\mathrm{d}S(a)}{\mathrm{d}a} \tag{9}$$

An **effective** benchmark is generally expected to establish a linear relationship between scores and model capabilities. However, when model ability falls significantly below average difficulty, the

binary scoring yields exponentially diminishing expected scores due to an extremely low correct rate. This results in exceptionally low discriminative power in such scenarios, rendering the benchmark ineffective at distinguishing model capabilities. Moreover, once a model's performance surpasses a certain threshold, its scores exhibit a remarkable improvement—a phenomenon that may lead researchers to misinterpret as the emergence of intrinsic model capabilities. To address such a problem, one possible method is to enlarge the difficulty variance, giving a more uniform difficulty distribution. Another effective method is to implement a partial correctness evaluation mechanism, such as the EED score, which significantly enhances both discrimination value and linearity in this region, offering higher information capacity. This mechanism is illustrated in Figure 8.

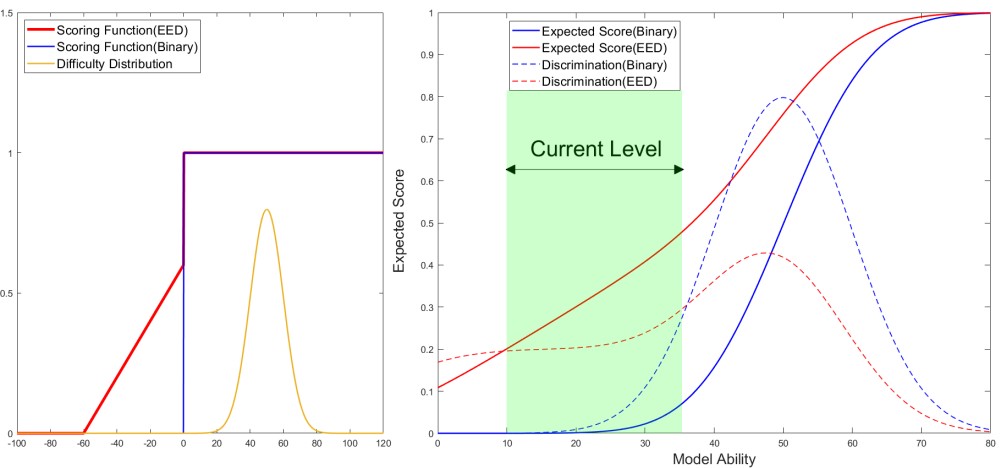

Figure 8: This figure qualitatively demonstrates the advantages of EED scoring over conventional binary scoring. Notably, in the lower score range, the EED scoring system exhibits a more linear relationship between final scores and model capabilities. The expected score is the convolution between the scoring function and the problem difficulty distribution function. Binary scoring results are drawn as red curves and our EED scoring results are drawn as red curves. Additionally, solid lines represent expected scores $S(a)$ while dashed lines indicate the discrimination $\frac{dS}{da}$ (i.e., the derivative of scores with respect to model capability).

The qualitative analysis above elucidates the rationale behind the EED Score's ability to assess model capability more precisely by quantifying structural dissimilarity between expressions. This theoretical insight is further supported by our empirical analysis presented in Appendix C.

## B.3   Limitations and Future Work of the EED Score

Although the EED Score succcessfully captures the detailed nuances between mathematical expressions as answers, it does not explicitly assess the correctness of the full reasoning process. While final-expression-based scoring enables efficient large-scale evaluation, it omits potentially important errors or reasoning flaws within intermediate steps. Prior work [22] shows that high-quality manual process-level evaluation is extremely resource-intensive and difficult to scale—typically limited to fewer than 10 problems for complex problems. Moreover, in physics, solution paths are often non-unique, making it challenging to define a single canonical trace for evaluation. This motivates our focus on end-result evaluation via symbolic expressions, but also highlights the need for more structured and scalable process-aware metrics.

Another improvement occurs during the calculation between tree structures where all the nodes are treated equally. In other words, it does not account for the physical plausibility of expressions such as dimensional correctness. One promising future direction is to augment symbolic edit-based metrics with physics-informed checks, such as unit analysis or symbolic dimensional validation. This could yield a more accurate assessment of physical reasoning beyond structural similarity.

Table 4: Performance of models on EED and accuracy metrics. Notation: $S_{\text{EED}}$ = EED Score; $\sigma_{\text{EED}}$ = EED Std Dev; $\text{CV}_{\text{EED}} = \sigma_{\text{EED}}/S_{\text{EED}} \times 100\%$ ; ACC = Accuracy; $\sigma_{\text{ACC}}$ = Accuracy Std Dev; $\text{CV}_{\text{ACC}} = \sigma_{\text{ACC}}/S_{\text{ACC}} \times 100\%$; Efficiency = $(\text{CV}_{\text{ACC}}/\text{CV}_{\text{EED}})^2$.

| Model | $S_{\text{EED}}$ | ACC | $\sigma_{\text{EED}}$ | $\sigma_{\text{ACC}}$ | CV$_{\text{EED}}$ (%) | CV$_{\text{ACC}}$ (%) | Efficiency |
|---|---|---|---|---|---|---|---|
| Gemini 2.5 Pro | 49.40 | 36.65 | 1.71 | 1.97 | 3.47 | 5.38 | 240.79% |
| o3 (high) | 46.30 | 34.58 | 1.72 | 1.91 | 3.71 | 5.53 | 221.48% |
| o4 mini (high) | 41.95 | 29.33 | 1.68 | 1.83 | 4.01 | 6.25 | 242.84% |
| DeepSeek-R1 | 37.78 | 24.88 | 1.59 | 1.71 | 4.20 | 6.87 | 267.24% |
| o3 mini (high) | 37.22 | 24.92 | 1.57 | 1.69 | 4.21 | 6.77 | 258.06% |
| o4 mini | 36.44 | 24.77 | 1.66 | 1.72 | 4.54 | 6.95 | 233.88% |
| o3 mini | 33.21 | 21.13 | 1.59 | 1.65 | 4.79 | 7.79 | 264.18% |
| Grok 3 Beta | 31.94 | 21.09 | 1.56 | 1.59 | 4.90 | 7.53 | 236.67% |
| Gemini 2.0 Flash Thinking | 30.25 | 17.93 | 1.48 | 1.51 | 4.88 | 8.40 | 296.31% |
| o1 | 27.46 | 10.72 | 2.03 | 1.27 | 7.40 | 11.86 | 257.09% |
| Claude 3.7 Sonnet Thinking | 27.12 | 15.25 | 1.44 | 1.43 | 5.30 | 9.40 | 314.68% |
| GPT-4.1 | 23.71 | 13.18 | 1.44 | 1.41 | 6.07 | 10.68 | 309.90% |
| DeepSeek-V3 | 24.17 | 13.45 | 1.39 | 1.38 | 5.75 | 10.27 | 318.79% |
| o3 mini (low) | 25.34 | 8.13 | 1.85 | 1.13 | 7.29 | 13.88 | 362.12% |
| Claude 3.7 Sonnet | 23.73 | 12.78 | 1.35 | 1.34 | 5.71 | 10.46 | 335.79% |
| GPT-4o | 15.35 | 6.89 | 1.11 | 1.04 | 7.26 | 15.12 | 434.02% |
| Qwen2.5-max | 13.92 | 6.03 | 1.04 | 0.96 | 7.44 | 15.83 | 452.20% |
| QwQ-32B | 4.54 | 1.58 | 0.94 | 0.51 | 20.77 | 32.26 | 241.21% |
| DeepSeek-R1-Distill-Qwen-32B | 3.19 | 0.70 | 0.71 | 0.35 | 22.30 | 49.56 | 493.72% |

Table 5: Pairwise Advantage Confidence. Each block is a confidence level of each row model outperforms the corresbonding column model. The OpenAI o-series is with reasoning effort="high".

| Model Model | Gemini 2.5 Pro | o3 | o4 mini | DeepSeek-R1 | o3 mini | GPT-4.1 | DeepSeek-V3 | GPT-4o |
|---|---|---|---|---|---|---|---|---|
| Gemini 2.5 Pro | 50% | 90% | 100% | 100% | 100% | 100% | 100% | 100% |
| o3 (high) | 10% | 50% | 96% | 100% | 100% | 100% | 100% | 100% |
| o4 mini (high) | 0% | 4% | 50% | 96% | 98% | 100% | 100% | 100% |
| DeepSeek-R1 | 0% | 0% | 4% | 50% | 60% | 100% | 100% | 100% |
| o3 mini (high) | 0% | 0% | 2% | 40% | 50% | 100% | 100% | 100% |
| GPT-4.1 | 0% | 0% | 0% | 0% | 0% | 50% | 41% | 100% |
| DeepSeek-V3 | 0% | 0% | 0% | 0% | 0% | 59% | 50% | 100% |
| GPT-4o | 0% | 0% | 0% | 0% | 0% | 0% | 0% | 50% |

## C Statistical Analysis

### C.1 Efficiency and Advantage Confidence

We employed a bootstrap analysis with 1000 resamples to evaluate the statistical uncertainty of our main results under the two metrics. The results are shown in Table 4. While the ranking of models remains consistent across both metrics, the EED Score demonstrate higher absolute values and smaller relative uncertainties compared to the accuracy metric. The relative uncertainty is proportional to the square root of sample size, allowing us to quantify the sample efficiency of the EED metric relative to the accuracy metric using the following formula:

$$\text{Sample Efficiency} = \left(\frac{\text{CV}_{\text{ACC}}}{\text{CV}_{\text{EED}}}\right)^2. \tag{10}$$

As shown in Table 4, our analysis reveals that the EED metric yields an average sample efficiency enhancement of 204% ($\sigma = 80\%$). This indicates that our benchmark under the EED metric with 500 problems provides evaluation strength equivalent to that under the accuracy metric with approximately 1500 problems, representing a substantial improvement in evaluation efficiency.

To establish the statistical significance of performance differences between models, we calculated pairwise advantage confidence levels. Using the scores and their associated uncertainties, we determined our confidence in asserting that one model outperforms another on PHYBench. The confidence level is calculated using Gaussian estimation:

$$\text{CL}_{s_i > s_j} = \Phi\left(\frac{\hat{s_i} - \hat{s_j}}{\sqrt{\sigma_{\hat{s_i}}^2 + \sigma_{\hat{s_j}}^2}}\right). \tag{11}$$

Notably, Gemini 2.5 Pro demonstrates superior performance with high confidence over most models, showing 99% confidence of outperforming all other models except o3 (90%). Table 5 also reveals clear performance tiers among the evaluated models, with statistically significant separations between the top performers (Gemini 2.5 Pro, o3 and o4 mini), mid-tier models (DeepSeek-R1, o3 mini), non-reasoning models (GPT-4.1, DeepSeek-V3) and legacy non-reasoning models (GPT-4o).

## C.2 Robustness Test on EED Scoring Metric

In this part, we show the robustness of EED scoring metric by changing its parameters, including its baseline score $s_0$, penalty coefficient $k$, and whether the subtree discount is enabled. The modified scoring function is defined as follows:

$$\text{score} = \begin{cases} 100, & \text{if } r = 0 \text{ (exactly match)}, \\ s_0 - kr, & 0 < r < \frac{s_0}{k}, \\ 0, & r > \frac{s_0}{k}. \end{cases} \tag{12}$$

Table 6: Rankings and Advantage Confidence of models under different parameters. Except for the last row, each cell in the table represents the change in the model's ranking under a specific baseline and penalty parameter setting compared to the configuration in the main text ($s = 60 - 100r$). The second column stands for model rankings under default scoring parameters. Column ACC stands for accuracy score. Column Conf represents the confidence level that each model performs better than the one ranked after it in PHYBench. The last row of the table shows the average sampling efficiency relative to ACC under the given parameter settings.

| Baseline
Penalty | 60,100
Ranking | Conf | ACC
ACC | 50
100 | 50
120 | 50
140 | 60
120 | 60
140 | 70
100 | 70
120 | 70
140 |
|---|---|---|---|---|---|---|---|---|---|---|---|
| Gemini 2.5 Pro | 1 | 93% | +0 | +0 | +0 | +0 | +0 | +0 | +0 | +0 | +0 |
| o3(high) | 2 | 91% | +0 | +0 | +0 | +0 | +0 | +0 | +0 | +0 | +0 |
| o4 mini(high) | 3 | 99% | +0 | +0 | +0 | +0 | +0 | +0 | +0 | +0 | +0 |
| DeepSeek-R1 | 4 | 56% | +1 | +0 | +1 | +1 | +0 | +0 | +0 | +0 | +0 |
| o3 mini(high) | 5 | 66% | -1 | +0 | -1 | -1 | +0 | +0 | +0 | +0 | +0 |
| o4 mini | 6 | 90% | +0 | +0 | +0 | +0 | +0 | +0 | +0 | +0 | +0 |
| o3 mini | 7 | 71% | +1 | +0 | +0 | +0 | +0 | +0 | +0 | +0 | +0 |
| Grok 3 Beta | 8 | 81% | -1 | +0 | +0 | +0 | +0 | +0 | +0 | +0 | +0 |
| Gemini 2.0 Flash Thinking | 9 | 64% | +1 | +0 | +0 | +1 | +0 | +0 | +0 | +0 | +0 |
| o1 | 10 | 83% | -1 | +0 | +0 | -1 | +0 | +0 | +0 | +0 | +0 |
| Claude 3.7 Sonnet Thinking | 11 | 78% | +0 | +0 | +0 | +0 | +0 | +0 | +0 | +0 | +0 |
| o3 mini(low) | 12 | 68% | +0 | +0 | +0 | +0 | +0 | +0 | +0 | +0 | +0 |
| DeepSeek-V3 | 13 | 56% | +0 | +0 | +0 | +1 | +0 | +0 | +0 | +0 | +0 |
| Claude 3.7 Sonnet | 14 | 54% | +1 | +1 | +1 | +1 | +0 | +1 | +0 | +0 | +0 |
| GPT-4.1 | 15 | 100% | -1 | -1 | -1 | -2 | +0 | -1 | +0 | +0 | +0 |
| GPT-4o | 16 | 83% | +0 | +0 | +0 | +0 | +0 | +0 | +0 | +0 | +0 |
| Qwen2.5-max | 17 | 100% | +0 | +0 | +0 | +0 | +0 | +0 | +0 | +0 | +0 |
| QwQ-32B | 18 | 86% | +0 | +0 | +0 | +0 | +0 | +0 | +0 | +0 | +0 |
| DeepSeek-R1-Distill-Qwen-32B | 19 | 0% | +0 | +0 | +0 | +0 | +0 | +0 | +0 | +0 | +0 |
| Average Efficiency | | 289% | 100% | 217% | 191% | 175% | 237% | 211% | 424% | 305% | 257% |

We report the variation in model rankings and sample efficiency under these settings in Table 6. Across most configurations, the rankings of the majority of models remain stable, with only minor fluctuations (within ±1 rank) observed for a few models. These fluctuations are largely attributable to low confidence margins (below 70%) in pairwise model comparisons. Additionally, enabling or disabling subtree discounting has no significant effect on overall ranking outcomes.

Regarding sampling efficiency, we observe that EED scoring methods exhibit significant improvements over the original ACC metric under variations of parameters. Although adopting a higher baseline score may appear to enhance sampling efficiency, this effect is merely an artifact of variance reduction caused by shifting non-perfect scores toward the full-score direction. These observations collectively demonstrate the robustness of our scoring methodology.

# D   Evaluation Experiment Setup

All models are queried with the following unified prompt template:

> You are a physics expert. Please read the following question and provide a step-by-step solution. Put your final answer, which must be a readable LaTeX formula, in a \\boxed{} environment.
>
> Question: {problem from PHYBench}
>
> Answer:

The final answer is then automatically extracted from within the \\boxed{} environment. We ignore any extra output outside the box, retain only the inner LaTeX expression, and tolerate additional text or commands inside the box as long as exactly one expression appears.

## E   TTS on Various Benchmarks

We selected some subsets of PHYBench and other baseline benchmarks for evaluation. For PHYBench, we chose the open source 100 questions; for AIME 2024[11], we used all 30 questions; and for OlympiadBench[10], MATH500 [13], and GPQA [24], we sampled 72 questions each. For OlympiadBench, we adopted 36 math problems and 36 physics problems, and among the physics problems we chose those labeled {"answer_type":"Expression"}.

Each benchmark uses the following unified prompt template:

> Please read the following question and provide a step-by-step solution. Put your final answer, which must be a readable LaTeX formula, in a \\boxed{} environment.{adapter}
>
> Question: {problem from PHYBench}
>
> Answer:

The contents of {adapter} vary across benchmarks:

- **PHYBench**, **OlympiadBench**: (empty)
- **GPQA**: Please answer with letter A, B, C, or D. (The final answer is extracted as the first uppercase letter inside the \boxed{} environment.)
- **AIME 2024**, **MATH500**: Please answer with a number.

Each model was evaluated 16 times per question. For certain smaller models, we conducted additional repetitions beyond 16 runs. In the graph, each data point corresponds to a sample pool size exceeding k, and a point is plotted only if over 90 percent of the questions were sampled more than k times. We plotted the pass@$k$ score (highest score among sampled answers, called accuracy) as a function of sampling size, along with the majority voting [28] score versus sampling size. During voting, equivalent expressions were treated as identical answers. We test both accuracy and EED Score.

### E.1   Pass@$k$

As the number of samples ($k$) increases during TTS, the model's capability does not grow indefinitely but instead approaches an upper bound. Due to budget constraints, the number of model responses we could test was limited. Therefore, we used an exponentially decaying curve to fit the model's capability boundary. The fitting formula employed was:

$$\text{Acc} = \text{Boundary} - \text{Gain} \cdot \exp\left(-\frac{x}{x_0}\right) \tag{13}$$

where Acc represents the accuracy or EED score, $x = \log k$ is the logarithmically transformed sampling count $k$ (with one sample corresponding to $x = 0$).Boundary, Gain, and $x_0$ are fitting parameters. Boundary is the upper bound.Gain represents the total Acc improvement achievable by increasing sampling, while $x_0$ denotes the decay rate toward the upper bound.

Table 7: Model Performance Boundaries on PHYBench under TTS.

| Model Name | pass@1 | pass@32 | vote32 | Boundary of pass@$k$ |
|---|---|---|---|---|
| Gemini 2.5 Pro | 38.71 | 65.91 | 41.97 | 74.9 |
| Gemini 2.5 Flash | 34.25 | 62.78 | 41.22 | 71.2 |
| DeepSeek-R1 | 25.06 | 50.88 | 28.65 | 81.3 |
| o4 mini | 23.2 | 52.1 | 24.6 | 78.6 |
| DeepSeek-V3 | 11.79 | 29.9 | 13.53 | not fitted |
| GPT-4o | 4.97 | 18.19 | 5.38 | not fitted |

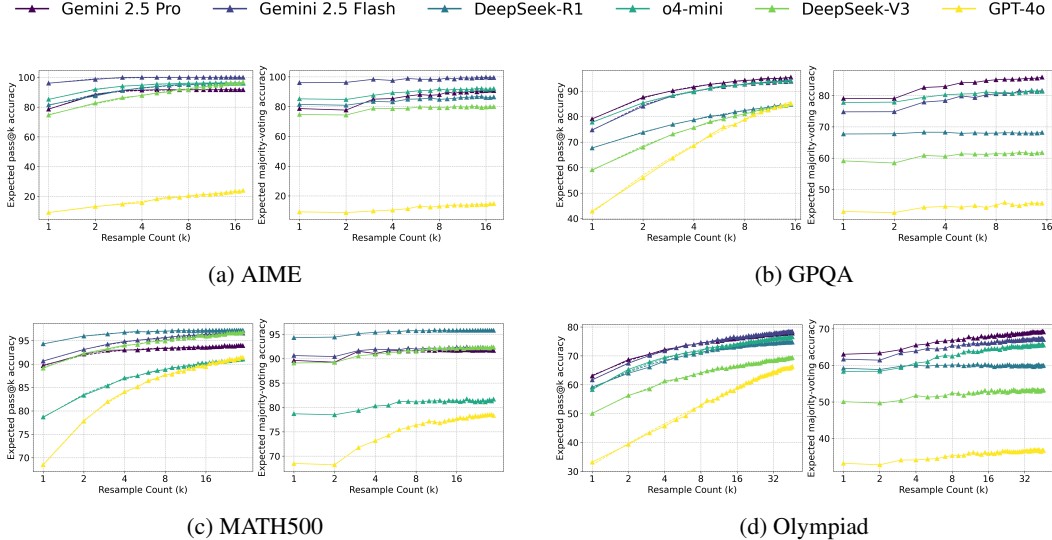

(a) AIME

(b) GPQA

(c) MATH500

(d) Olympiad

Figure 9: Combined metrics comparison across different datasets. For each dataset, the left figure shows the pass@$k$ results and the right figure shows the majority voting results

The results for each benchmark, including pass@$k$ EED score, pass@$k$ accuracy, majority voting EED score, and majority voting accuracy, are shown in Figure 9. The fitted curve (dashed line) was applied only to the pass@$k$ data. The x-axis represents the logarithmically transformed sampling count, and the y-axis represents the accuracy or EED score. For PHYBench, the pass@$k$ results are shown in Figure 10.

The fitting results reveal two findings: (1) the curve fitted by exponential decay aligns well with our data, indicating that its upper bound is also credible; (2) the curves for lower-scoring language models exhibit a notably linear trend. The fitting results of A, B, C are shown in Table 7.

## E.2 Majority Voting

As shown in Figure 11, majority voting provides only a modest improvement in accuracy on PHY-Bench, typically by a few percentage points. This limited gain suggests that while models can generate diverse outputs, their ability to select the correct one remains weak. In contrast, the pass@$k$ strategy leads to significantly larger improvements—often exceeding dozens of points—across both reasoning and non-reasoning models. This indicates that correct answers do exist in the model's output space, but models struggle to recognize them. Together, these results highlight a key bottleneck: current models possess some capacity for reasoning but lack reliable self-evaluation mechanisms.

## F  Illustrative Case Studies of PP and RR Errors

This section provides a detailed demonstration of the reasoning process behind PP and RR. We outline their definitions and roles within typical solution traces, and present concrete case studies illustrating

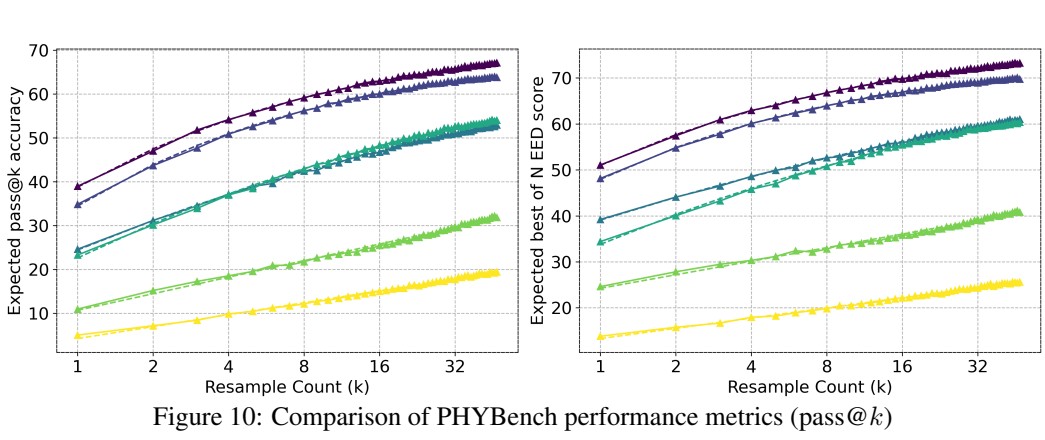

Figure 10: Comparison of PHYBench performance metrics (pass@$k$)

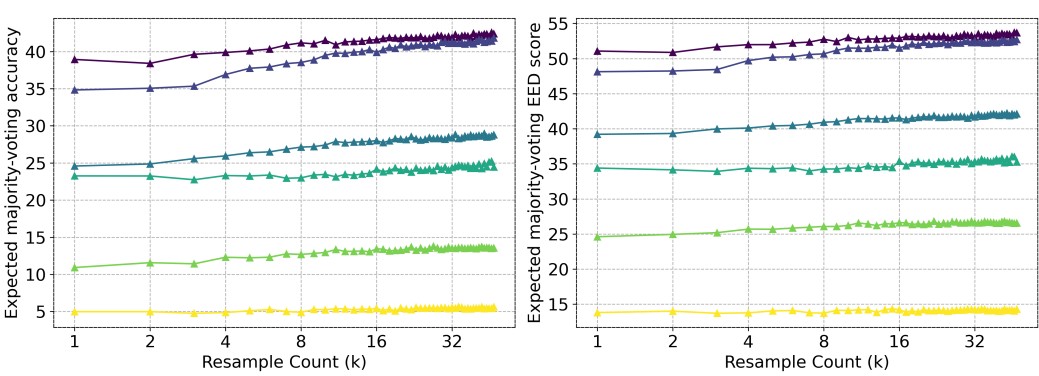

Figure 11: Comparison of PHYBench performance metrics (majority voting)

how representative models fail in each category. These examples highlight the characteristic structure of PP and RR, and clarify how specific errors—such as incorrect physical modeling or inconsistent derivation—can lead to failure.

## F.1  Illustration of PP and RR Process

**Example Reasoning Process**

**Physical Perception (PP):**
First, I need to understand the entire system's initial state and ... I should draw a sketch. ... the tension is continuous, but I still have to analyse each ball's forces one by one. ... the strings haven't had time to swing yet. The top ball's sudden horizontal motion requires centripetal force ...

**Robust Reasoning (RR):**
From equation (3):
$$T_3 - mg = ma_{1r}$$
so
$$T_3 = mg + ma_{1r}$$
Substitute into equation (2):
$$T_2 - (mg + ma_{1r}) - mg = ma_{1r}$$

which becomes

$$T_2 - mg - ma_{1r} - mg = ma_{1r}$$

...

Substitute the expression for $T_2$:

$$T_1 = (2mg + 2ma_{1r}) + mg + ma_{1r} = 3mg + 3ma_{1r} \cdots$$

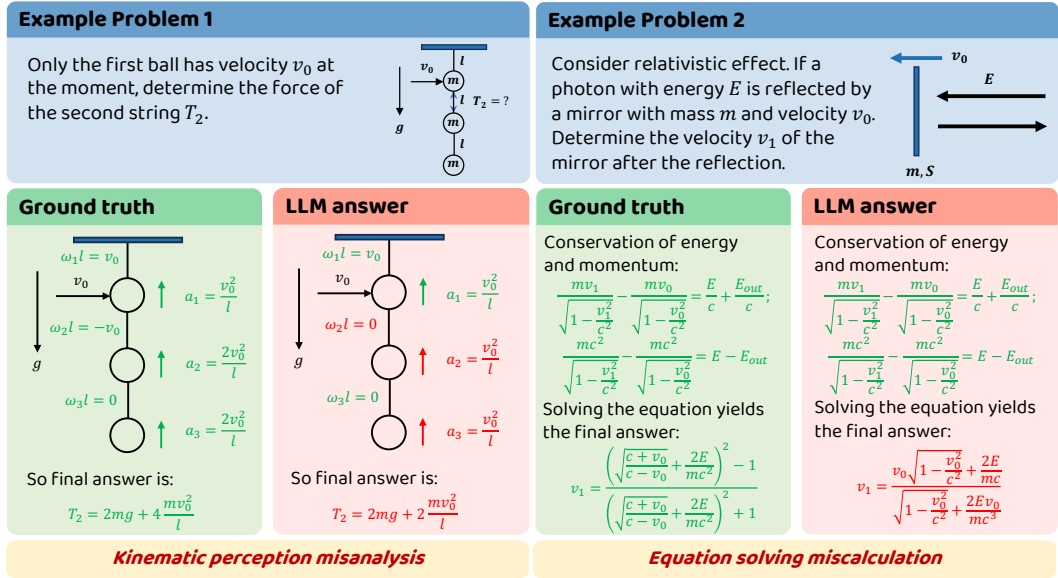

Figure 12: Example questions and errors from the solution generated by DeepSeek-R1. Here we demonstrate the main parameters and physical processes. See Appendix H for the full question.

As discussed in Section 5, from a structural perspective, PP represents decision nodes while RR forms the connecting links in the reasoning chain. Errors at PP nodes can lead to fundamental misunderstandings of the physical scenario, resulting in incorrect answers. They may also introduce unnecessary physical effects, complicating subsequent symbolic reasoning. Meanwhile, RR errors involve inconsistencies in deriving expressions, solving equations, or applying conditions, which accumulate and cause the final expression to increasingly diverge from the correct answer.

### F.2 Case Study of PP

The first typical challenge arises from an insufficient understanding of physical processes and inadequate modeling skills. As illustrated in Figure 12, **Example Problem 1** presents a classical mechanics scenario involving three balls connected by an inextensible string. The erroneous solution from the LLM results from a misunderstanding of the kinematics relationships among these balls, perceiving the angular velocity of the middle string to be zero incorrectly. Even if the symbolic derivation is right, the model results in a wrong answer.

The PP challenge in this problem is easy for average college students, but even cutting-edge models like Gemini 2.5 Pro, o3 and DeepSeek-R1 failed to handle this kinematics. Our experiments further reveal that 32B models perform especially poorly on PP phases, often failing even on elementary problems. Such failures highlight not only a fundamental limitation in the models' perception capacity but also semantic reasoning.

### F.3 Case Study of RR

Another common error involves maintaining consistency across lengthy and intricate reasoning processes, as well as difficulties in accurately solving the resulting equations. For instance, in

Figure 12, **Example Problem 2** presents a scenario where a mirror, moving at relativistic speed, is recoiled by a high-energy photon. Although the LLM correctly interpreted the physical setup and identified the appropriate equations, it ultimately failed to derive the correct solution after an extended symbolic derivation. This reflects a typical lack of robustness in mathematical reasoning.

Physics problems often require extensive symbolic manipulation. Due to space limitations, the two illustrative problems shown are relatively short; however, as noted earlier, the average length of a full solution in PHYBench is approximately 3,000 characters, and human solvers typically employ dozens of intermediate expressions before arriving at the final answer. Moreover, when unaided by external mathematical tools, LLMs tend to generate significantly more intermediate steps than human reference solutions, bringing more risks of making mistakes. This observation suggests that physics problems effectively represent long-range reasoning tasks constrained by diverse but definite rules. Our experimental results indicate that such long-range symbolic reasoning remains a significant challenge for current models.

# G   Chain-of-Thought Poisoning Protocol

PHYBench problems demand long-range, step-wise reasoning in which each step contains key symbolic expressions that can be verified. This property makes PHYBench an ideal testbed for evaluating the robustness of reasoning and even probing whether LLMs' reasoning is **genuine** or **superficial**. In this section, we provide a detailed implementation of our perturbation experiment.

## G.1   Experimental Settings

For every target model we evaluate eight perturbation conditions (two baselines + six toxins) as follows:

1. Select an PHYBench problem and truncate its reference solution.
2. Inject one systematic perturbation from the catalogue in Appendix G.2.
3. Submit the dialogue [`prompt` → `poisoned CoT` → "continue"] with the template in Appendix G.5, and record whether the model detects or propagates the error.

## G.2   Perturbation Catalogue

Each perturbation keeps the original problem statement intact but appends either a faithful or a corrupted partial solution. The canonical quantity being tampered with is $(R_m + h)^2$.

**F1. Baseline with raw problem** The original problem.

**F2. Baseline with correct partial solution** The problem is augmented with the unperturbed partial solution. This baseline is tested to test whether partial reasoning effects model accuracy.

**T1. Remove square term** The square in the law of gravitation is removed.

$$(R_m + h)^2 \rightarrow (R_m + h)$$

The flaw is not obvious in later algebra but can be exposed by dimensional analysis.

**T2. Operator reversal** Replaces the plus sign with a minus, introducing a physically implausible expression:

$$(R_m + h)^2 \rightarrow (R_m - h)^2$$

**T3. Combine T1 and T2** Applies both T1 and T2 simultaneously to examine compound error sensitivity:

$$(R_m + h)^2 \rightarrow (R_m - h)$$

The presence of two conflicting cues increased the probability that many models flagged an error, raising their aggregate scores.

**T4. Remove $h$ term** Deletes the $h$ term while keeping the square, mimicking a subtle simplification:

$$(R_m + h)^2 \rightarrow R_m^2$$

Although dimensionally sound, this error is conceptually obvious; e.g. DeepSeek-R1's accuracy plummeted from 97 % to 37 % under this perturbation.

**T5. Combine T2 and T4** A compound error similar to T3, used as a control:

$$(R_m + h)^2 \rightarrow R_m$$

Functions as a counterpart to T3, allowing for comparative analysis of compound perturbations.

**T6. Alter physical law** Rewrites the hydrostatic equilibrium equation, flipping the physical dependency:

$$\frac{dP}{dh} = -\alpha \, g(h) \rightarrow \frac{dP}{dh} = -\frac{\alpha}{g(h)}$$

Tests whether the model recognises blatant violations of fundamental physical laws.

## G.3 Illustration of Superficial Reasoning and Genuine Reasoning

---

**The partial solution of Remove square term (T1)**

To derive the expression for the temperature $T(h)$ of the martian atmosphere as a function of height $h$, we will follow these steps:

**Step 1: Understand the Given Density Profile** The mass density $\alpha(h)$ is given by:

$$\alpha(h) = \rho_0 \left(1 + \frac{h}{R_m}\right)^{1-n}$$

where $\rho_0$ is a constant, and $n > 4$.

**Step 2: Relate Density to Pressure** For an ideal gas, the pressure $P$ is related to the density $\alpha$ and temperature $T$ by:

$$P = \frac{\alpha}{\mu} RT$$

where $\mu$ is the molar mass of the atmosphere, and $R$ is the ideal gas constant.

**Step 3: Hydrostatic Equilibrium** The pressure gradient in the atmosphere is balanced by gravity. The hydrostatic equilibrium equation is:

$$\frac{dP}{dh} = -\alpha g(h)$$

where $g(h)$ is the gravitational acceleration at height $h$.
The gravitational acceleration $g(h)$ at height $h$ is:

$$g(h) = \frac{GM_m}{(R_m + h)} \quad \rightarrow \quad \text{it should be} (R_m + h)^2$$

**Step 4: Substitute and Differentiate** Substitute $P$ from Step 2 into the hydrostatic equilibrium equation:

$$\frac{d}{dh}\left(\frac{\alpha RT}{\mu}\right) = -\alpha \frac{GM_m}{(R_m + h)} \quad \rightarrow \quad \text{it should be} (R_m + h)^2$$

Assuming $T$ is a function of $h$, we can expand the derivative:

$$\frac{R}{\mu}\left(\frac{d\alpha}{dh}T + \alpha\frac{dT}{dh}\right) = -\alpha\frac{GM_m}{(R_m + h)} \quad \rightarrow \quad \text{it should be} (R_m + h)^2$$

---

**Superficial Reasoning: propagate the error without reflection**

Continuing from the derivative expression:

$$\frac{R}{\mu}\left(\frac{d\alpha}{dh}T + \alpha\frac{dT}{dh}\right) = -\alpha\frac{GM_m}{(R_m + h)}$$

---

**Step 5: Compute $\frac{d\alpha}{dh}$ From the given density profile**

$$\alpha(h) = \rho_0 \left(1 + \frac{h}{R_m}\right)^{1-n}$$

, we compute the derivative with respect to $h$:

$$\frac{d\alpha}{dh} = \rho_0(1-n)\left(1 + \frac{h}{R_m}\right)^{-n} \cdot \frac{1}{R_m}$$

Substitute

$$\alpha = \rho_0 \left(1 + \frac{h}{R_m}\right)^{1-n}$$

:

$$\frac{d\alpha}{dh} = \frac{(1-n)}{R_m}\left(1 + \frac{h}{R_m}\right)^{-1}\alpha$$

**Step 6: Substitute $\frac{d\alpha}{dh}$ Back** Substitute $\frac{d\alpha}{dh}$ into the derivative equation:

$$\frac{R}{\mu}\left(\frac{(1-n)}{R_m}\left(1 + \frac{h}{R_m}\right)^{-1}\alpha T + \alpha\frac{dT}{dh}\right) = -\alpha\frac{GM_m}{(R_m + h)}$$

Divide through by $\alpha\frac{R}{\mu}$:

$$\frac{(1-n)}{R_m}\left(1 + \frac{h}{R_m}\right)^{-1}T + \frac{dT}{dh} = -\frac{GM_m\mu}{R(R_m + h)}$$

**...**

**Verification:** If we substitute

$$T(h) = \text{constant}$$

back into the hydrostatic equilibrium equation, we find consistency, confirming that the temperature is height-independent for the given density profile.

**Final Answer:**

$$\boxed{T(h) = \frac{GM_m\mu}{R(n-1)}}$$

---

**Genuine Reasoning: correct the answer through semantic understanding**

But wait, the gravitational acceleration on Mars at a height $h$ would be $g(h) = GM_m/(R_m + h)^2$, right? Because gravity decreases with the square of the distance from the center. But in the previous assistant's step 3, they wrote $g(h)$ as $GM_m/(R_m + h)$. Hmm, that seems incorrect. Wait, gravitational acceleration is $GM$ divided by $r$ squared. So if $R_m$ is the radius of Mars, then at height $h$, the distance from the center is $R_m + h$, so $g(h) = GM_m/(R_m + h)^2$. But the assistant wrote it as $GM_m/(R_m + h)$. That must be a mistake. Wait, maybe I should check that. Let me think. The gravitational acceleration at the surface is $g_0 = GM_m/R_m^2$. At height $h$, it's $g(h) = GM_m/(R_m + h)^2$. So the assistant's expression for $g(h)$ is wrong. That might affect the subsequent steps. But the user's question says to continue from the previous answer, which had $g(h) = GM_m/(R_m + h)$. So maybe there's a typo here. Wait, but the assistant's step 3 says "gravitational acceleration $g(h)$ at height $h$ is $GM_m/(R_m + h)$". That's missing a square in the denominator. That's a mistake. But since the user is asking to continue from where the assistant left off, perhaps I need to follow that, but that would lead to an incorrect result. Hmm, this is a problem. Wait, maybe the assistant made a mistake here, and I need to correct it. But the user wants me to continue from the previous steps. Let me check the previous steps again.

Okay, let me try to work through this problem step by step. $\cdots$ Wait, perhaps the problem assumes that the gravitational acceleration varies as $1/(R_m + h)$, but that's not standard. Normally, gravitational acceleration decreases with the square of the distance. So this might be a key point. Wait, but maybe the problem is using a different approximation. For example, if $h$ is much smaller than $R_m$, then $R_m + h \approx R_m$, so $g \approx GM_m/R_m^2$. But the problem, the density is given as a function that includes $(1 + h/R_m)$, which suggests that $h$ could be a significant fraction of $R_m$. But the problem states that the atmosphere is very thin, so maybe $h$ is not that large. Hmm. Alternatively, perhaps the problem is using a different form for some reason. Wait, the assistant's Step 3 says $g(h) = GM_m/(R_m + h)$. That's dimensionally correct if $GM_m$ has units of $m^3/s^2$, but $R_m + h$ is in meters, so $g$ would have units $m^2/s^2$, which is not correct. Wait, no, gravitational acceleration has units of $m/s^2$. So $GM_m/(R_m + h)$ would have units $(m^3/s^2 \cdot kg)/m$ ? Wait, no, $GM_m$ has units of $(m^3/s^2)$, since $G$ is in $m^3 kg^{-1} s^{-2}$, multiplied by $M_m$ (kg) gives $m^3/s^2$. Then dividing by $(R_m + h)$ (meters) gives $m^2/s^2$. Which is not acceleration. So that's a mistake. Therefore, the assistant made an error in Step 3 by writing $g(h)$ as $GM_m/(R_m + h)$. The correct expression should be $GM_m/(R_m + h)^2$. So this is a critical error. But since the user is asking to continue from the previous steps, which include this mistake, then the assistant's solution would be wrong. But the user is asking for the correct answer. So perhaps the problem statement actually has a different form for $g(h)$. Let me check the original problem again. Wait, the problem says: "the mass of Mars is $M_m$ (far greater than the total mass of the martian atmosphere), and its radius is $R_m$." So the gravitational acceleration at height $h$ would be $g(h) = GM_m/(R_m + h)^2$. So the assistant's Step 3 is incorrect. Therefore, the assistant's Step 4 is using the wrong $g(h)$. Therefore, the following steps would be incorrect.

## G.4 Original Problem

**Problem 719 (THERMODYNAMICS)**

The Martian atmosphere can be considered as composed only of very thin $CO_2$. The molar mass of this atmosphere is denoted by $\mu$, and the atmosphere at the same height can be considered as an ideal gas in equilibrium. The mass of Mars is $M_m$ (far greater than the total mass of the Martian atmosphere), and its radius is $R_m$. Assume the relationship between the atmospheric mass density and the height $h$ above the surface of Mars is given by

$$\alpha(h) = \rho_0 \left(1 + \frac{h}{R_m}\right)^{1-n},$$

where $\rho_0$ is a constant, and $n\,(n > 4)$ is also a constant. Derive the expression for the temperature $T(h)$ of the martian atmosphere as a function of the height $h$. Express the result using the following physical quantities: the mass of Mars $M_m$, radius $R_m$, molar mass of the atmosphere $\mu$, constant $n$, gravitational constant $G$, and ideal gas constant $R$.

**Answer:**

$$\boxed{T(h) = \frac{\mu\, G\, M_m}{n\, R\, (R_m + h)}}$$

**Unperturbed Chain-of-Thought Summary:**

- To derive the expression for the temperature $T(h)$ of the martian atmosphere as a function of height $h$, we will follow these steps:
- **Step 1: Understand the Given Density Profile** The mass density $\alpha(h)$ is given by:

$$\alpha(h) = \rho_0 \left(1 + \frac{h}{R_m}\right)^{1-n}$$

  where $\rho_0$ is a constant, and $n > 4$.
- **Step 2: Relate Density to Pressure** For an ideal gas, the pressure $P$ is related to the density $\alpha$ and temperature $T$ by:

$$P = \frac{\alpha}{\mu} RT$$

where $\mu$ is the molar mass of the atmosphere, and $R$ is the ideal gas constant.

- **Step 3: Hydrostatic Equilibrium** The pressure gradient in the atmosphere is balanced by gravity. The hydrostatic equilibrium equation is:

$$\frac{dP}{dh} = -\alpha \, g(h)$$

where $g(h)$ is the gravitational acceleration at height $h$.

The gravitational acceleration $g(h)$ at height $h$ is:

$$g(h) = \frac{GM_m}{(R_m + h)^2}$$

- **Step 4: Substitute and Differentiate** Substitute $P$ from Step 2 into the hydrostatic equilibrium equation:

$$\frac{d}{dh}\left(\frac{\alpha RT}{\mu}\right) = -\alpha \frac{GM_m}{(R_m + h)^2}$$

Assuming $T$ is a function of $h$, we can expand the derivative:

$$\frac{R}{\mu}\left(\frac{d\alpha}{dh}T + \alpha\frac{dT}{dh}\right) = -\alpha \frac{GM_m}{(R_m + h)^2}$$

- **Step 5: Compute $\frac{d\alpha}{dh}$** Differentiate $\alpha(h)$ with respect to $h$:

$$\frac{d\alpha}{dh} = \rho_0(1-n)\left(1 + \frac{h}{R_m}\right)^{-n} \cdot \frac{1}{R_m}$$

Substitute into the hydrostatic equation:

$$\frac{R}{\mu}\left(\frac{\rho_0(1-n)}{R_m}\left(1 + \frac{h}{R_m}\right)^{-n}T + \rho_0\left(1 + \frac{h}{R_m}\right)^{1-n}\frac{dT}{dh}\right)$$

$$= -\rho_0\left(1 + \frac{h}{R_m}\right)^{1-n}\frac{GM_m}{(R_m + h)^2}$$

Simplify:

$$\frac{R}{\mu}\left(\frac{(1-n)}{R_m + h}T + \frac{dT}{dh}\right) = -\frac{GM_m}{(R_m + h)^2}$$

- **Step 6: Solve for $\frac{dT}{dh}$** Multiply through by $\frac{\mu}{R}$:

$$\frac{(1-n)}{R_m + h}T + \frac{dT}{dh} = -\frac{\mu GM_m}{R(R_m + h)^2}$$

- **Step 7: Integrating Factor** This is a linear ODE of the form:

$$\frac{dT}{dh} + P(h)T = Q(h), \quad P(h) = \frac{1-n}{R_m + h}, \quad Q(h) = -\frac{\mu GM_m}{R(R_m + h)^2}$$

The integrating factor is:

$$I(h) = e^{\int P(h)\,dh} = (R_m + h)^{1-n}$$

Multiply through:

$$(R_m + h)^{1-n}\frac{dT}{dh} + (1-n)(R_m + h)^{-n}T = -\frac{\mu GM_m}{R}(R_m + h)^{-n-1}$$

Recognize the LHS as the derivative of a product:

$$\frac{d}{dh}\left(T(h)\,(R_m + h)^{1-n}\right) = -\frac{\mu GM_m}{R}(R_m + h)^{-n-1}$$

- **Step 8: Integrate Both Sides**

$$T(h)\,(R_m + h)^{1-n} = -\frac{\mu GM_m}{R}\int (R_m + h)^{-n-1}\,dh + C$$

$$T(h)\,(R_m + h)^{1-n} = \frac{\mu GM_m}{nR}(R_m + h)^{-n} + C$$

$$T(h) = \frac{\mu GM_m}{nR}(R_m + h)^{-1} + C(R_m + h)^{n-1}$$

- **Step 9: Boundary Condition** Require $T(h) \to$ finite as $h \to \infty$, so $C = 0$ (since $n > 4$). Final result:

$$T(h) = \frac{\mu GM_m}{nR}\cdot\frac{1}{R_m + h}$$

Or equivalently:

$$T(h) = \frac{\mu GM_m}{nRR_m}\left(1 + \frac{h}{R_m}\right)^{-1}$$

### G.5 Implementation Prompt Template

We present the prompt template used for all perturbation experiments. The full dialogue, including the system and user messages, is shown below.

```
prompt = ("Please read the following question and provide a step-by-step "
          "solution. Put your final answer (LaTeX) inside \boxed{}.\n\n"
          f"Question: {problem['content']}\n\nAnswer:")

messages = [
    {"role": "user",      "content": prompt},
    {"role": "assistant", "content": poisoned_cot},    # T1-T6 variant
    {"role": "user",      "content":
     "Please **continue** from your previous reasoning. "
     "Do NOT restart from Step 1."}
]
```

## H   Example Questions

### H.1   Full Question Text for Given Errors in Figure 12

**Example Problem 1:** Three small balls are connected in series with three light strings to form a line, and the end of one of the strings is hung from the ceiling. The strings are non-extensible, with a length of $l$, and the mass of each small ball is $m$. Initially, the system is stationary and vertical. A hammer strikes one of the small balls in a horizontal direction, causing the ball to acquire an instantaneous velocity of $v_0$. Determine the instantaneous tension in the middle string when the topmost ball is struck. (The gravitational acceleration is $g$.)

**Example Problem 2:** Consider an ideal mirror moving at relativistic velocity, with mass $m$ and area $S$. (The direction of photon incidence is the same as the direction of the mirror's motion.) Now consider the case where the mirror is moving with an initial velocity $\beta_0 c$. In this situation, the mirror is unconstrained by external forces, and photons are incident on it with constant power for a certain period of time, with energy $E$. Assuming the mirror's velocity after irradiation is $\beta_1 c$, find the expression for $\beta_1$.

### H.2   Demonstration of Selected Problems

We demonstrate 5 additional problems with their answers. For more detailed information, please refer to the PHYBench website.

## Selected Problem 1

A smooth bowl with a radius of $R$ is fixed, and the plane at the mouth of the bowl is horizontal. A smooth, homogeneous, thin rod $AB$ with length $L = \frac{4\sqrt{3}R}{3}$. B is located outside the bowl, while end A presses against a point inside the bowl. The rod achieves static equilibrium in a plane passing through the center of the sphere $O$. Points $D$ and $D'$ on the rod are nearly coincident with the point of contact at the rim of the bowl, but $D$ is slightly lower-left, and $D'$ is slightly upper-right. Let the angle between the rod and the horizontal plane be $\theta$. The rod is suddenly cut at point $D$. Note that after being cut, point $D$ will gently rest on the inner surface of the bowl. Find the angular acceleration $\beta = \ddot{\theta}$ of the rod at this instant.

**Answer:**

$$\beta = -\frac{g}{2R}$$

## Selected Problem 2

Consider a child with mass $m$ sitting on a swing, the child can be regarded as a point mass with the mass concentrated at the seat plank. Ignore the mass of the other parts of the system. The distance from the swing seat plank to the pivot is $l$. At this time, consider the frictional torque $M_f = a$ (where $a$ is a constant) at the swing's suspension point. There is someone behind who applies an impulsive torque $J_0$ to the swing every time it reaches the furthest back position. Find the difference in speed rates $\Delta v$ of the child after passing the lowest point twice successively when the motion reaches a steady state (with gravitational acceleration $g$ and assuming the swing angle is relatively small).

**Answer:**

$$\Delta v = \sqrt{gl\left(\frac{J_0{}^2}{8aml^2} + \frac{a}{mgl}\right)}\left(\sqrt{\frac{J_0{}^2}{8aml^2} + \frac{3a}{mgl}} - \sqrt{\frac{J_0{}^2}{8aml^2} - \frac{a}{mgl}}\right)$$

## Selected Problem 3

Consider an infinite-length black body with inner and outer cylinders, which are in contact with heat sources at temperatures $T_1$ and $T_2$, respectively; assume that the temperature of the heat sources remains constant. Let the inner cylinder have a radius $r$, the outer cylinder have a radius $R$, and the distance between the axes of the inner and outer cylinders be $b$, with $r < b < R$ and $r + b < R$. Find the power $p(\theta)$ absorbed per unit area from the heat source at angle $\theta$ on the surface of the outer cylinder (i.e., the power density at $\theta$), where $\theta$ is the angle between the line connecting a point on the surface of the outer cylinder and the center of the outer cylinder, and the line connecting the centers of the inner and outer cylinders. The Stefan-Boltzmann constant is denoted as $\sigma$.

**Answer:**

$$p(\theta) = (\sigma T_2^4 - \sigma T_1^4)\frac{r(R - b\cos\theta)}{R^2 + b^2 - 2Rb\cos\theta}$$

## Selected Problem 4

A square loop with side length $a$ and mass $m$ is made from a resistive material, with a total resistance of $R$. At $t = 0$, the loop is located at $x = 0$ and moves with a velocity $v_0\hat{x}$. The loop lies in the $x$-$y$ plane. There is a magnetic field $\mathbf{B} = B_0\left(\frac{x}{x_0}\right)\hat{z}$, where $B_0 > 0$ is a constant. In this problem, we ignore the effects of gravity. What is the velocity $v(t)$ of the square loop at time $t$? Write the expression for $v(t)$ in terms of $t$ using the parameters $B_0$, $v_0$, $a$, $m$, and $R$.

**Answer:**

$$v(t) = v_0 e^{-\frac{1}{mR}\left(\frac{a^2 B_0}{x_0}\right)^2 t}$$

**Selected Problem 5**

For the electromagnetic cannon model, its structure consists of two parallel rails spaced $l$ apart, with one end connected to a power supply for energy, and the other end connected to a metal rod that can slide freely on the rails to form a circuit. In the situation where the circuit length $x$ is much larger than the spacing $l$ (but ignoring the delay in circuit signal propagation caused by the length), it can be assumed that the self-inductance coefficient $L$ of the circuit is linearly related to $x$, i.e., $L = Ax + B$. $A$ and $B$ are two constants. The current flowing through the metal rod is $I$, and the permeability of vacuum is $\mu_0$. In fact, for different electromagnetic cannon configurations, the value of the Ampere force on the metal rod is actually different. Assume the rail is a thin-walled cylinder with a radius $r \ll l$. Under direct current conditions, it can be assumed that the current is uniformly distributed over the surface of the cylinder. Make an appropriate approximation and calculate the specific expression of the Ampere force on the metal rod.

**Answer:**

$$\frac{\mu_0 I^2}{2\pi} \ln \frac{l}{r}$$

