# OpenReview forum: "PHYBench: Holistic Evaluation of Physical Perception and Reasoning in Large Language Models"
_NeurIPS.cc/2025/Datasets_and_Benchmarks_Track — NeurIPS 2025 Datasets and Benchmarks Track poster_

### Official Review · Reviewer_EsZ9 · 2025-06-01

**Rating:** 4
**Confidence:** 4

**Summary:**

This work: (1) proposes a new dataset and benchmark, PhyBench, with 500 high difficulty Physics Olympiad level problems that are carefully created and checked by college students from School of Physics (most with Physics Olympiad experience and most are gold medal (first prize) winner in CPhO), and provide human experts' baselines; (2) propose the EED score, a rule-based evaluation metric that is more fine-grained and can improve sample efficiency while evaluation; and (3) evaluate LLMs on the proposed datasets, showing significant gap between LLMs and human experts. They evaluate reasoning models and non-reasoning models, and study the pass@k performance of each model; the authors also carry out detailed error analysis on where and why the models fail.

**Additional Feedback:**

I recommend the authors to think or act in the following directions. Here are my suggestions,

(1) **How to strictly generate new Physics Olympiad problems according to an existing Physics Olympiad problem, with purely rule-based methods, and with describeable 'distance'?** For example, modifiying to a different extent would lead to different difficulty distance of the modified problem. Take problems in your dataset as examples, problem id 726 is only restating the Envelope of Parabolic Motion, which does not increase its difficulty; while problem (id 702, ELECTRICITY, "To establish a rectangular...") seems to be modifying a classic problem from Physics Olympiads, and has indeed increased its difficulty. **Based on such rule-based modification methods**, you might modify each problem when testing some certain Language Model. This would increase variance, but would avoid data leakage in a better way that is truly suitable for Physics Olympaid problems.

(2) Add EED score, and make better existing datasets like OlympiadBench[1]. This is a more labor-efficient approach, and hence can provide more problems.

(3) Avoid over-claiming.

**Personally, I appreciate your hard work on curating such dataset:** **I was a gold medalist in Physics Olympiads** years earlier in high school myself, and **I know how tiring (or, labor-consuming) it is to solve Olympiad problems**. I am very interested in evaluating or improving LLMs' performance on Physics Olympiads, and **I believe your resources have the potential to provide really interesting datasets and findings to the community in the future.**

References:

[1] OlympiadBench: A Challenging Benchmark for Promoting AGI with Olympiad-Level Bilingual Multimodal Scientific Problems, Chaoqun He et al. https://arxiv.org/abs/2402.14008.

[2] Investigating Data Contamination for Pre-training Language Models, Minhao Jiang et al. https://arxiv.org/abs/2401.06059

**Dataset Code Accessibility:**

Partly

**Dataset Code Comments:**

On huggingface the authors have provided all 500 problems, 100 of which are with publicly available solutions and answers. The code authors provided in the supplementary material is about calculating EED scores and is correct. Though the authors have not made all solutions to problems available, I personally think this is totally reasonable to avoid potential data leakage caused by training on the evaluation benchmark. Therefore, **I think there is sufficient detail to support reproducibility, though the authors have not yet made public answers to all problems (to avoid potential data leakage)**. I appreciate the authors' actions, and this part does not lower my ratings.

**Ethical Considerations:**

No, there are no or only very minor ethics concerns

**Final Justification:**

The authors have carried out plenty of experiments during the discussion period regarding: (1) my main concern about contributions, (2) my concern about how much better is the new dataset. While my concern about contribution remains not solved, the authors have shown successfully that previous Physics Olympiad datasets are indeed lacking rigorous checking to a certain extent.

However, my main contribution regarding contributions remains not resolved. 500 problems are not big for a dataset, and the level of novelty of each problem were overclaimed in the first version of the author's submission. There do exist Level-1 novelty problems, and I am not convinced by the authors' proposed method for evaluating novelty. If this dataset is composed of low-level novelty problems, then I don't think it is of good quality. That's why I still recommend rejection till now.

I'd like to further clarrify that I am not decreasing my score becasue of human labor in this work, and I am not criticizing the authors for using such human labor. I feel a bit pitty that, perhaps through some more labor-efficient method, like modifying existing datasets and carefully checking existing datasets, the authors could have provided the community with a better dataset with much more problems.

**Limitations Weaknesses:**

The reasons why I recommend rejection are as described:

The authors **are reinventing the wheel again compared to previous work, while not improving existing datasets well enough; moreover, there is over claiming in the paper**. Comparing to previous work: I don't see huge improvement compared to OlympiadBench[1], where they have also included 8000 Olympiad problems, including 2000 Physics problems from global and regional Physics Olympiad. **Authors claim improvements in the following aspects:** (a) proposing EED scores of higher sample efficiency, (b) seemingly more strict human-based evaluation pipelines and higher quality, and (c) `entirely of original problems' to avoid data leakage.

**However,**

(a) The authors only provide 500 problems, which is much smaller than 2000 problems of OlympiadBench. Is EED improving sample efficiency enough to let 500 samples beat 2000 samples? Also, why not simply add EED onto existing benchmarks, which can be much more labor-efficient?

(b) OlympiadBench[1] uses problems from IPhO, APhO, EPhO, USAPhO, PUPC, OPhO: all are very formal and carefully checked Physics Olympiads with high quality problems. The claim authors made in Appendix A seems to be questionable. For example, authors claim that Problem 1015 has a wrong gt answer and lack explanation of $\gamma$, but $\gamma$ is mentioned in it, and **the gt answer is correct, while the author-claiming 'Actually correct' model answer is wrong**. I have checked this problem myself. (I was a gold medalist in Physics Olympiads years earlier when I was in high school.)

(c). (c.1.) The term 'entirely of original' is not true and is over claiming; (c.2.) The claim that your dataset avoid data leakage compared to previous datasets is not validated on Language Models; (c.3.) The dataset proposed actually cannot avoid data leakage problem.

(c.1.) **The terms 'entirely of original', 'totally original', etc., (which apper in Section 1, in Section 6, in Checklist problem 11, etc.)** **are over claiming**. For example, the problem (id 726, MECHANICS, "A cannon located on the ground...") is simply about Envelope of Parabolic Motion, which is very common, either in General Physics textbooks or in Physics Olympiads. **Obviously General Physics textbooks are easier to find compared to Physics Olympiad problems in IPhO, hence the probability that LLMs trained by large companies have seen this problem is even higher than the probability that they have seen IPhO problems.** Also, some of the problems might be modified, but only to a little extent, for example problem (id 285, MECHANICS, "A small bug with a mass of $m$...") might be a modified problems from some other problems, but still very similar to some existing problems. I personally do not think simply restating a common problem can be called 'entirely original'.

(c.2.) The claim that your dataset avoids pretrain data leakage **is not validated with Language Models**. The authors have not used any pretrain data leakage detection methods (e.g. those described in [2]) to detect and compare data leakage in the proposed PhyBench dataset and existing datasets.

(c.3.) Even you have shown currently your benchmark performs better than others on data leakage with some data leakage detection methods, a fixed dataset still cannot avoid data leakage or overfitting, **especially when you are not using entirely original problems**. Actually I think the authors are not utilizing the property of Physics Olympiad problems; many Olympiad problems are modular, and can be modified based on rule-based methods (to avoid data leakage). I will provide some suggestions later in my review.

**To conclude: this work is not improving previous datasets to an enough extent, and is overclaiming part of its advantage.**

**Strengths Contributions:**

The strength of this work is obvious. (1) The authors use great human effort to collect and check a new dataset; (2) the authors proposed and ablate a new evaluation metric, EED, for improving sample efficiency; (3) The authors carry out detailed analysis on how language models fail on the proposed dataset.

The authors claim their proposed dataset (1.1) is difficult enough for evaluating sota LLMs, (1.2) can avoid potential data contamination, and (1.3) is of higher quality compared to existing benchmarks.

---

> ### Author Rebuttal · Authors · 2025-07-30
>
> **Response to Reviewer EsZ9**
>
> We appreciate your examination of our work and expertise as a **Physics Olympiad gold medalist**. As physics competition enthusiasts and medalists ourselves, we value your professional peer review.
>
> Before addressing the specific points, we hope to clarify the guiding philosophy behind PHYBench, which we believe addresses many of the points raised. Our benchmark's primary contribution is its rigorous evaluation standard: **every problem is designed and manually verified to be fairly scored by the equivalence of the final symbolic expression**. This strict format is our core quality guarantee, designed to eliminate the ambiguities inherent in other evaluation paradigms and fundamentally reduce scoring errors.
>
> We believe strict design prevents evaluation noise that obscures a model's true reasoning capabilities. Specifically, directly using Olympiad problems for evaluation creates deficiencies in handling answer diversity (e.g., equivalent expressions with different parameters, applications of small quantity approximations) and flexible scoring. We will illustrate this using the IPhO example you raised.
>
> This demonstrates that PHYBench is not simply reinventing the wheel, but rather **a benchmark with substantial contributions and innovations**.
>
> **Q(a)**: The authors only provide 500 problems, much smaller than OlympiadBench's 2000 problems. Does EED improve sample efficiency enough for 500 samples to beat 2000 samples? Why not simply add EED to existing benchmarks, which would be more labor-efficient?
>
> **A(a)**: You raise a fair point about dataset size. Our 500 problems are quantitatively fewer than 2000, even with EED Score's ~204% efficiency gain (equivalent power of ~1500 binary-scored problems). However, this scale sufficiently differentiates models. Our results in Figure 1 clearly shows the performance gap between current LLMs.
>
> Regarding your second question about applying EED to other benchmarks: we can do this, but only for benchmarks with expression-based answers. Moreover, the primary value of PHYBench is the meticulously curated dataset itself, whose rigorous design cannot be substituted by the sheer quantity from other sources.
>
> **Q(b)**: OlympiadBench uses problems from IPhO, APhO, EPhO, USAPhO, PUPC, OPhO: all are very formal and carefully checked Physics Olympiads with high quality problems. The claim authors made in Appendix A seems to be questionable. For example, authors claim that Problem 1015 has a wrong gt answer and lack explanation of γ, but γ is mentioned in it, and the gt answer is correct, while the author-claiming 'Actually correct' model answer is wrong.
>
> **A(b)**: Let us first address Problem 1015 specifically (from IPhO 2011 Problem 2, subproblem 4). We apologize for incorrectly stating γ was undefined. It was indeed defined. However, we maintain the Model Answer is correct for the following reasons:
> 1. The problem explicitly asks for answers "in terms of ρs, R0, g, t and η", while γ is not included in this list.
> 2. Subproblem 2.3 already calculated the γ effect to be on the order of **0.01%**. This strongly suggests it should be omitted as a negligible correction.
> 3. Omitting γ aligns with both the problem statement and physical intuition developed throughout the problem.
>
> We consulted the **2011 Chinese IPhO team coach and two IPhO gold medalists**. They confirmed that both answers would receive full marks in actual grading. However, automated metrics cannot replicate this human flexibility.
> This example illustrates a broader issue with directly using Olympiad problems for AI evaluation. While competitions like IPhO provide high-quality problems, they are designed for human contestants with flexible grading practices. The diversity in scoring methods and answer expressions constitutes evaluation noise that obscures true model capabilities. PHYBench addresses this by requiring single-expression answers and using multiple rounds of human calibration to ensure each problem is self-contained. Specifically, we establish that every problem can be uniquely scored through its expression format, with human review guaranteeing this standard. This eliminates evaluation noise and provides clear, unambiguous standards needed for rigorous AI assessment.
>
> **Q(c)**: Regarding originality claims - (c.1) The term 'entirely of original' is not true and is over claiming; (c.2) The claim that your dataset avoid data leakage compared to previous datasets is not validated on Language Models; (c.3) The dataset proposed actually cannot avoid data leakage problem.
>
> **A(c.1)**: We sincerely appreciate your detailed examination of PHYBench problems. We acknowledge that Problems 726 and 285 are based on classic physics models without sufficient modification to demonstrate originality. Moreover, we contacted the contributors of Problem 702. They adapted it from the 38th CPhO Problem 4 by transforming a current-carrying wire's magnetic field to a charged wire's electric field, maintaining the same geometric configuration with only rotation and scaling. They then solved the new problem independently to provide an original solution.
>
> We firmly believe the majority of PHYBench problems are either originally created by our contributors or substantially adapted from unpublished sources not available online. We will revise our claims throughout the paper, changing "entirely original" to "originally curated or carefully modified" to more accurately reflect our dataset composition.
>
> **A(c.2)**: You rightly pointed out that our initial submission lacked formal validation of its low contamination risk. While the methods in the paper you cited ([2]) study the phenomenon of contamination without providing a universal detection tool for unknown pre-training sets, we have now conducted two reproducible experiments to address your concern. We are willing to provide additional details in the discussion.
>
> **Experiment 1: LLM-based "Novelty Score"** - To quantify the novelty of our dataset, we prompted leading LLMs (GPT-4o, Gemini-2.5, Qwen-Max) and a search-augmented model (Deepseek-R1-Search) to act as "Novelty Judgers." They were asked to score problems from PHYBench and other benchmarks on a scale of 0-100 based on the likelihood of the exact question or a minor variation appearing in public web data.
>
> | Novelty Judger | PHYBench | GPQA Diamond | MMMU | OlympiadBench | PhysUniBench |
> |----------------|----------|--------------|------|---------------|--------------|
> | GPT-4o | **75.25 ± 0.75** | 65.29 ± 0.63 | 65.70 ± 0.45 | 61.11 ± 0.51 | 64.14 ± 1.24 |
> | Gemini-2.5 | **60.33 ± 1.62** | 47.23 ± 0.51 | 24.63 ± 0.61 | 37.90 ± 0.68 | 21.56 ± 1.90 |
> | Qwen-Max | **83.64 ± 0.24** | 58.24 ± 1.08 | 48.69 ± 0.62 | 71.86 ± 0.74 | 62.82 ± 0.97 |
> | Deepseek-R1-search | **41.87 ± 1.86** | 26.91 ± 1.46 | 19.46 ± 0.67 | 22.71 ± 0.80 | 16.86 ± 0.70 |
>
> *Table: Mean novelty scores. Higher is better.*
>
> PHYBench **consistently ranks the highest**, showing that it is not likely to be drawn from well-known public sources or have been discussed. Notably, the more recently published PhysUniBench, which partially relies on web-scraped data, shows markedly lower novelty than PHYBench.
>
> **Experiment 2: Perturbation via Narrative Rewriting** - To test for overfitting on surface-level phrasing, we used GPT-4.1 to generate paraphrased versions of our problems. The core physics, mathematical structure, and final answer were preserved, but the narrative background was significantly altered. We then evaluated several LLMs on these rewritten problems.
>
> | Model             | Original | Rewritten_1 | Rewritten_2 | Rewritten_3 |
> |-------------------|----------|-------------|-------------|-------------|
> | gemini-2.5-flash  | 45.08    | 39.33       | 43.44       | 44.84       |
> | gpt-4.1           | 32.62    | 29.63       | 32.42       | 28.41       |
> | o4 mini           | 42.86    | 32.57       | 43.90       | 39.98       |
>
> *Table: Mean score before or after rephrasing. Less drop is better.*
>
> The minimal performance drop indicates that the model is not simply relying on memorizing a specific question but is engaging with the underlying physics. This is exactly what we want to evaluate.
>
> **A(c.3)**: We agree that complete originality is nearly impossible for physics problems. We will clarify our claims to emphasize that our problems are either originally created or substantially adapted, not "totally original." While our approach is resource-intensive, requiring significant human effort, it effectively minimizes data leakage compared to using publicly available problems.
>
> We are particularly intrigued by your suggestion of modular, rule-based modification methods for physics problems. If you could provide more specific implementation details or examples, we would be eager to explore this approach for properly utilizing physics problem structures in future work. Your expertise would be invaluable in developing such methods.
>
> **Response to Additional Feedback:**
>
> Firstly, rule-based generation is an interesting direction, but current models cannot reliably solve existing physics problems, making synthetic variations difficult to validate. Additionally, EED Score cannot be applied to most current benchmarks because they focus on numerical answers rather than expression-based responses. Moreover, we thank you for pointing out our overclaiming. We have revised all instances of "entirely original" throughout the paper and appreciate your correction.
>
> Finally, we're grateful for your recognition of our work. PHYBench advances the field through: **self-contained problems with unambiguous answers** compared to existing benchmarks like OlympiadBench; innovative evaluation via EED Score for expression-based assessment; **human baseline**; and revealing insights about model reasoning patterns. We believe these contributions represent considerable innovation and respectfully hope you will consider our responses in your final evaluation.

---

> > ### Comment · Reviewer_EsZ9 · 2025-08-02
> > **EED score: problem 244 and problem 600 in your dataset also have value-based answers; many Physics problems have expression-based answers**
> >
> > My main concern is still about novelty, contribution and overclaiming, while it seems EED score won't raise an exception for value-based answers (e.g.problem 244 or 600 in your dataset). Moreover, many Physics Olympiad problems have expression-based answers.

---

> > ### Author Response · Authors · 2025-08-06
> > **Response Regarding Numerical vs Expression-based Questions**
> >
> > The difference between PHYBench and OlympiadBench is not the format, but the more diverse problems and a more rigorous screening for evaluability. The EED Score can be applied to the expression-based parts of OlympiadBench, but the core issue is the accuracy of the evaluation metric—a deficiency that simply adopting OlympiadBench cannot rectify.

---

> ### Comment · Reviewer_EsZ9 · 2025-08-02
> **Thanks for the response, still concerned about novelty, contribution and overclaiming**
>
> I'd like to thank the authors for their response.
>
> Actually, my main concern is **still about contribution and over-claiming**. Let me first explain what 'rule-based generation' and 'describe-able distance' I referred to mean, and how this relates to my evaluation of your work's contribution and potential over-claiming. I think the importance here is that **you have to provide quantitatively how novel your problems are according to some standard**.
>
> First, you have to come up with some standard or 'distance' your problems are. I appreciate your trials on the experiments with novelty scores of LLMs. As a domain expert in Physics Olympiad, for your reference, I personally provide a level how 'novel' a physics problems is, **compared to its 'nearest neighbor' in existing Physics Olympiad problems available**:
>
> 1. **Not Novel**. The physics problem is almost the same (even in narrative level) with some existing problem.
>
> 2. **Level 0.5. Narrative Rewriting**. For example, letting LLMs generate re-phrased versions of the problem; changing some notations, etc. Answers remain the same or very much unchanged.
>
> 3. **Level 1. Small Perturbation**, similar to the idea of Math-P-Simple in [1]. The Problem Solving Method remains almost the same, and math work remains almost the same. For example, perhaps Problems 726 and 285.
>
> 4. **Level 2. Same or Similar Physics model**, more complicated Math, or different Physics knowledge, perhaps similar idea to Math-P-hard in [1]. For example, from single-Dof oscillator to multi-Dof oscillator, etc.. For example, Problem 702.
>
> 5. **Level 3. New Physics model** (of some existing problem solving method). For example, perhaps the first Olympiad Problem considering using Maxwell Equations for birefringence (if not considering existing papers about this).
>
> 6. **Level 4. New Physis problem-solving method**. This would be quite hard. I agree that it is almost impossible to come up with many Level-4 novel problems.
>
> **This standard is my first thought and might be imperfect. However, again, it's important that you have to show quantitatively through similar standard how novel your problems really are.**
>
> The perturbation via Narrative Rewriting experiment only proves that your problems are at least in Level 0.5: However, simply using LLMs to generate rewritten problems can achieve this level.
>
> Say, following this standard, how many of your problems are in Level 0.5, Level 1 and Level 2, correspondingly?
>
> **The original claims in your paper (entirely original, etc.) make readers think that all your problems are at least in Level 2, which is not the case.**
>
> **As for 'originally curated or carefully modified': I appreciate the huge amount of work you have done for creating such a dataset, however, if your dataset is mainly consisted of Level 0.5 or Level 1 problems, I don't think you are using a labour-efficient approach, and I don't think this is of high novelty nor contribution.** (For example, I appreciate authors' explanation with respect to Problem 1015; though model's answer to this problem (if provided alone) should not be considered correct, I know in IPhO the marking standard would be less strict, and this is indeed a good example of why existing datasets need modification to some extent. **However, generating a dataset with Level 0.5 or Level 1 with correct problems perhaps only needs sota LLMs and much less human effort than mentioned in this work, and does not have enough contribution**). Please show in methods like I mentioned how novel your dataset is.
>
> **As for the LLM-judged novelty, personally I don't think this is very reliable on judging novelty, since this looks quite a vibe approach. Could you refer to the levels I mentioned and conduct similar classification of your problems, which makes more sense?**
>
> Another note is about data contamination: as far as I know, many of the Large LLM companies (like OpenAI, Anthropic, Qwen, Seed) are actually using quite powerful internet spider to search for and even buy high standard Olympiad problems, hence some problems like those in CPhO, or even those in out-of-school education and training organizations, might be viewed as being data-contaminated.
>
> [1] MATH-Perturb: Benchmarking LLMs' Math Reasoning Abilities against Hard Perturbations

---

> > ### Comment · Reviewer_EsZ9 · 2025-08-02
> > **More explanation about rule-based modification**
> >
> > What I referred to as rule-based means without human-in-the-loop or AI-in-the-loop. For example, we can generate 100% correct arithmetic problems about sum of integers purely with rule-based methods. Say, can you think of method to generate Level 0.5, Level 1 or even higher Level Pertubation Olympiad problems with rule-based methods? This would be a fun direction to think of (but indeed looks quite far beyond what we have now).

---

> > > ### Author Response · Authors · 2025-08-06
> > > **Response Regarding Rule-Based Modification**
> > >
> > > Thanks for raising this point. We appreciate the opportunity to discuss this relevant technology, but rule-based modification is **orthogonal to our core contribution—the benchmark itself**.
> > >
> > > Regarding the mathPerturb work you mentioned, that study relied on manual perturbations by human experts and targeted ordinary math problems where models already achieved 90% accuracy. For physics problems, perturbation would likely be considerably more challenging. GSM-Symbolic converted numerical values to symbols and established computational graphs for programmatic perturbation, but this approach only works effectively for a limited subset of problems.
> > >
> > > We confirm that high-level, fully automated modification is still far from practical implementation, at least in physics. If achieved, most STEM datasets could be augmented for evaluation or training purposes, representing an entirely separate contribution. **PHYBench already provides a suitable evaluation platform for assessing current LLMs' physics reasoning capabilities.**

---

> > ### Author Response · Authors · 2025-08-03
> > **Regarding contribution of PHYBench**
> >
> > Firstly, we have to address your concerns about our contribution at a high level. PHYBench problems demonstrate exceptional difficulty and quality—Gemini 2.5 Pro achieves only 37% accuracy, while GPT-4o scores merely 7%. Crucially, all problems are human-verified as solvable, meaning sufficiently capable AI could approach nearly full scores. We believe this alone represents a **significant contribution** and should **alleviate data contamination concerns**.
> >
> > While we cannot guarantee every problem is unseen by all models, the vast majority are at least "unsolved". You mentioned that "Large LLM companies" may have included most internet and commercial problems in training, yet their scores remain low. This demonstrates PHYBench problems are either genuinely novel or resistant even to overfitting. We achieved this without filtering out problems current models can already solve.
> >
> > PHYBench effectively differentiates current models' physics reasoning abilities with scores far from saturation (ensuring continued relevance). Previous physics benchmarks fail to distinguish advanced models—for instance, on OlympiadBench, DeepSeek-R1 in January, Gemini 2.5 Pro, and o4-mini in April achieve nearly the same scores. Do you truly believe models haven't progressed in physics reasoning? **PHYBench clearly captures the progress of advanced reasoning models**.
> >
> > You repeatedly mention our approach isn't "labor efficient," suggesting we could build a good dataset using existing physics benchmarks with rule-based perturbations and EED Score. We respectfully disagree. You acknowledge our strength in human curation—this is crucial. Our reviewer system ensures high evaluation metric accuracy, an optimization absent in previous programmatically-extracted benchmarks.  Existing datasets like OlympiadBench and UGPhysics would still require human curation to achieve this standard. However, their homogeneous sources (PhOs or "The Great Compendium of Physics Problems" from the University of Science and Technology of China) could advantage models trained on these specific materials. Therefore, we chose to have students directly create diverse, evaluable problems. We sincerely recommend you to reconsider this fundamental principle and its value to the community.

---

> > ### Author Response · Authors · 2025-08-03
> > **Regarding originality and novelty of PHYBench**
> >
> > We appreciate your proposed framework for examining how physics questions differ from their "original base". However, the definition of "original base" requires clarification. The "original base" may refer to all searchable online problems or training datasets of "Large LLM companies," this would be meaningful but practically impossible—no company provides access to their training corpora.  Given these constraints, we conducted an empirical evaluation using your framework. We prompted DeepSeek-Search to assess our problems' adaptation levels using your definitions directly. The results demonstrate that **the majority of PHYBench problems fall into Level 2 and Level 3**, indicating substantial modification from any potential source material.
> >
> > | Benchmark | Sample Size | Level 0.5 | Level 1 | Level 2 | Level 3 | Level 4 |
> > | :--- | :---: | :---: | :---: | :---: | :---: | :---: |
> > | PHYBench | 100 | 0 | 9 | 44 | 47 | 0 |
> > | OlympiadBench | 376 | 36 | 261 | 65 | 14 | 0 |
> > | PhysUniBench | 50 | 4 | 32 | 9 | 5 | 0 |
> >
> >
> > Here's the prompt for this experiment. The results are averaged over 5 trials.
> >
> >
> > ```python
> > _prompt_template_intro: str = """
> > **You are an expert AI tasked with evaluating the intellectual novelty and originality of research topics or questions. Your objective is to assess how unique and unexplored a given topic is, based *exclusively* on the provided internet search results and the original topic itself.
> >
> > **Your Goal:** Assign a Novelty Score (0.5,1,2,3 or 4) and provide a detailed justification.
> >
> > **Inputs You Will Receive:**
> >
> > 1.  **`[ORIGINAL_TOPIC_TEXT]`**: The specific topic or question you need to evaluate.
> >
> > 2.  **`[INTERNET_SEARCH_RESULTS_TEXT]`**: A compilation of text snippets, titles, summaries, or full articles obtained by searching the internet for `[ORIGINAL_TOPIC_TEXT]`. This represents the existing body of knowledge found online.
> >
> > **Novelty Score Definition (0.5,1,2,3 or 4):**
> >
> > - Level 0.5. Narrative Rewriting. For example, letting LLMs generate re-phrased versions of the problem; changing some notations, etc. Answers remain the same or very much unchanged.
> > - Level 1. Small Perturbation, similar to the idea of Math-P-Simple (These perturbations, such as changing numbers or asking slightly different questions, maintain the same solution approach as the original problem.). The Problem Solving Method remains almost the same, and math work remains almost the same.
> > - Level 2. Same or Similar Physics model, more complicated Math, or different Physics knowledge, perhaps similar idea to Math-P-hard (These perturbations significantly change the nature of the problem, making the original solution method inapplicable.). For example, from single-Dof oscillator to multi-Dof oscillator, etc..
> > - Level 3. New Physics model (of some existing problem solving method). For example, perhaps the first Olympiad Problem considering using Maxwell Equations for birefringence (if not considering existing papers about this).
> > - Level 4. New Physis problem-solving method. This would be quite hard.
> >
> > """
> >
> > _prompt_template_instruction: str = """
> > Please evaluate the following question and answer from the perspective of *memorization likelihood* and be precise in your judgment.
> >
> > You must base your judgment on whether this QA pair resembles content that is publicly accessible on the internet (e.g., educational websites, open datasets, online forums, exam banks, research papers, etc.).
> >
> > **Avoid relying on intuition or general reasoning alone**. Instead, imagine whether this QA pair could be found—word-for-word or with slight edits—on sites like **Google**, Wikipedia, StackExchange, arXiv, Khan Academy, or other searchable online sources.
> >
> > Assess both:
> > - The novelty of the question phrasing and structure.
> > - The originality of the explanation or solution process.
> >
> > Only assign a high novelty score if the content seems clearly unfamiliar, custom-constructed, or stylistically distinct from common web-based materials.
> >
> > Now, here are the QUESTION and the ANSWER:
> > """
> >
> > _prompt_template_format: str = """
> > Bear in mind the content above. Ponder how novel and original the QApair is and give your answer.
> >
> > ABSOLUTELY FOLLOW THESE FORMAT REQUIREMENTS:
> > - Your response must EXACTLY match this format, character for character
> > - Do NOT include any markdown symbols like ** or *
> > - Do NOT add any extra text outside the specified format
> > - The score must be a number 0.5,1,2,3 or 4
> >
> > Required format:
> > Novelty Score: <score 0.5,1,2,3 or 4>
> > Explanation: <your concise reasoning>
> >
> > Example valid responses:
> > Novelty Score: 2
> > Explanation: xxx
> >
> > WARNING: Our parsing script uses this exact pattern to extract the score:
> > match = re.search(r"Novelty Score[:：]?\\s*([0-9]+(?:\\.[0-9]+)?)", answer)
> > Any deviation from the required format will cause parsing failure.
> > """
> >
> > full_prompt = (
> >     _prompt_template_intro + "\n\n" +
> >     _prompt_template_instruction + "\n\n" +
> >     "QUESTION:\n" + question + "\n\n"+
> >     _prompt_template_format
> > )
> > ```

---

> > ### Author Response · Authors · 2025-08-03
> > **Regarding overclaiming**
> >
> > We have revised all instances of "entirely original" to "originally curated or carefully modified" throughout the paper. The novelty analysis above should address your remaining concerns about the originality of our problems. With this clarification, we maintain that the paper's title and core idea, PHYBench as a "Holistic Evaluation of Physical Perception and Reasoning in Large Language Models," remain an accurate description of our contribution.

---

> > ### Author Response · Authors · 2025-08-03
> > **Regarding data contamination**
> >
> > We claimed in our Introduction that the three challenges for existing reasoning benchmarks are over-simplication, data contamination and lack of accuracy in evaluation metrics. You have committed that we fixed the third challenge. The first two challenges, in our opinion, can be validated by the low performance of current cutting-edge models.
> >
> > If you were to ask about and check our methodology, we addressed these issues by having a large number of humans create problems, with reviewers instructed to reject overly conventional questions. We deliberately included some conventional problems, and the standards for rejection varied across time, thereby enhancing the diversity of our problem distribution. Our data sources are diverse, unlike previous benchmarks such as OlympiadBench and UGPhysics, which relied on relatively uniform sources, or contemporary benchmarks like PhysUniBench and PHYSICS, which depend on internet-sourced data. Our dataset is therefore **less likely to be contaminated**.

---

> ### Comment · Reviewer_EsZ9 · 2025-08-03
> **Thanks the authors for their response, my concern regarding contribution remains unsolved**
>
> I would like to thank the authors for the keen efforts made in trying to address my concerns; however, a strict, rigorous analysis of novelty of each problem is missing, hence my concern remains.
>
> The purely LLM-scored novelty is unreliable for me: you are not using RAG systems that enables LLMs to compare against existing problems.
>
> I'd also like to emphasize that I am not criticizing or down-scoring the huge amont of efforts you have made; but rather, **I am particularly concerned about to-what-extent your dataset is better than a dataset obtained by using LLMs to narratively rewrite existing Physics Olympiad datasets** (i.e. dataset composed of Level 0.5 novel problems).
>
> Since this main concern of mine remains unaddressed, I cannot recommend acceptance(till now). I am happy to increase my score to 3: boarderline reject in the end of the discussion period **in response to your efforts made in rebuttal**. However, **my main concern about contribution and overclaiming remains unaddressed**, and **I would like to share my comments and this concern about contribution and overclaiming with other reviewers and ACs (please see my review and previous discussions with authors).** I believe they would make comprehensive and appropriate evaluations.
>
> I'm also open for further discussions. If you could provide strict, rigorous analysis of novelty of each problem in your dataset (as we have discussed), I would consider to mark my concern as being addressed.

---

> > ### Author Response · Authors · 2025-08-03
> >
> > We appreciate your emphasis on rigorous novelty assessment and share the desire for quantitative validation where feasible. We agree that an **explainable, quantitative** analysis of novelty is ideal.
> >
> > However, we must address a fundamental obstacle: a comprehensive reference corpus of all existing physics problems, the prerequisite for a RAG-based analysis, is unavailable. This renders a problem-by-problem "strict novelty proof" methodologically intractable for any research group. This is likely why most benchmarks establish novelty via transparent provenance and community scrutiny, a standard we have followed.
> >
> > We are also inspired by your proposed levels to generate a more quantitative estimate from search-augmented models, simulating the retrieval of similar problems from the Internet and model's implicit knowledge base.. Our intention with the analysis was not to present a flawless, exhaustive proof. Rather, it was a principled, good-faith effort to operationalize your insightful framework within the bounds of what is scientifically feasible. **If you could kindly suggest a specific reference dataset for comparison as well as an evaluation metric for novelty, we would be more than willing to conduct a targeted quantitative analysis.**
> >
> > We wish to address your concern with another quantitative metric. Since we lack a definitive reference corpus for direct RAG-based comparison, a seemingly clear quantitative indicator of our benchmark's novelty and difficulty is **model performance itself**.
> >
> > Despite being publicly available for over three months, the best score on PHYBench has only improved from 37% to 42%, achieved by Grok 4. We believe that Grok and Gemini, both strong in physics, have encountered extensive physics problems during their training, yet they still solve less than half of our problems. If our problems were simply rewritten versions of existing ones, we would expect much higher performance from these leading models. If you find this quantitative metric insufficient, we may not have better alternatives available.
> >
> > Ultimately, we believe the combination of clear data sources, community review, our framework-based novelty estimate, and consistently low performance by top models provides strong evidence of PHYBench's contribution. We remain committed to further analysis and would be happy to conduct a targeted quantitative comparison if a specific and appropriate reference dataset is suggested.
> >
> > We sincerely appreciate your recognition and willingness to increase the score. More importantly, we are grateful for your exceptionally thorough engagement with our work. This level of diligent and constructive feedback is essential for upholding the integrity of our research community.

---

> > ### Author Response · Authors · 2025-08-06
> > **Kind Follow-up on Our Discussion**
> >
> > We hope this message finds you well. We wanted to kindly follow up on our previous response regarding the novelty assessment methodology, as we haven't yet received your thoughts on the points we raised.
> >
> > We remain genuinely enthusiastic about implementing your valuable suggestions for a more rigorous quantitative analysis. As we mentioned, we are fully prepared to conduct the targeted experiments you envision. Any instructions or advice you might offer—perhaps regarding suitable reference datasets, evaluation metrics, or experimental protocols—would be helpful as we work to strengthen our validation approach.
> >
> > **We fully understand you have many responsibilities and deeply appreciate the time you've already invested in reviewing our work.** Your expertise and insights have been invaluable, and we would be grateful for any further guidance you feel comfortable providing.
> >
> > Looking forward to continuing our discussion at your convenience.

---

### Official Review · Reviewer_ncFb · 2025-06-30

**Rating:** 4
**Confidence:** 1

**Summary:**

This paper presents PHYBench, a new benchmark designed to evaluate the physical reasoning abilities of LLMs. The benchmark consists of 500 original physics problems, ranging from high school to Olympiad difficulty, and introduces the Expression Edit Distance Score for more granular assessment of model outputs. While the work is well-executed and addresses key gaps in existing benchmarks, there exisst some limitations.

**Additional Feedback:**

- Are there plans or methods to expand the benchmark beyond physics?
- The EED score evaluates final answers but not intermediate reasoning. Did the authors consider incorporating step-wise correctness checks? Are there some  challenges?
- Does the tree edit distance algorithm face scalability issues for complex expressions (e.g., high-order equations with many variables)? Have you tested its runtime performance or explored optimization methods (e.g., approximation or parallelization) for large-scale computations?

**Dataset Code Accessibility:**

Yes

**Dataset Code Comments:**

The provided code is functional and well-structured, facilitating reproducibility. However, some scripts could be more extensively documented to aid users in adapting the benchmark for different use cases.

**Ethical Considerations:**

No, there are no or only very minor ethics concerns

**Limitations Weaknesses:**

- The benchmark primarily targets Olympiad-level physics, which may limit its applicability to more general or advanced reasoning tasks.

- While the EED Score effectively assesses final answers, it does not verify the correctness of intermediate reasoning steps, a critical aspect of true problem-solving.

**Strengths Contributions:**

- The problems are carefully curated to avoid data contamination, and the validation process ensures high-quality, unambiguous questions.

- The proposed EED Score provides a more nuanced way to assess symbolic reasoning compared to traditional binary scoring.
- The paper thoroughly examines where and why LLMs fail, offering valuable insights into model limitations.

---

> ### Author Rebuttal · Authors · 2025-07-29
>
> **Response to Reviewer ncFb**
>
> Thank you for your thoughtful review and constructive feedback. We address the two main concerns below:
>
> **Q1: The benchmark primarily targets Olympiad-level physics, which may limit its applicability to more general or advanced reasoning tasks.**
>
> **A1:** We respectfully but strongly disagree that PHYBench is limited in applicability. We acknowledge the current emphasis on Olympiad-level problems and are committed to expanding PHYBench across multiple difficulty tiers—from high school to frontier research level. However, we clarify that our current 500 problems **already span from high school to undergraduate difficulty levels**, as briefly mentioned in our introduction.
>
> Our current dataset provides **exceptional diagnostic value** through three key contributions: (1) the first human-curated benchmark specifically designed for rigorous evaluation of complex physical reasoning; (2) the EED Score that provides fine-grained assessment with **204% improved sample efficiency**; and (3) revealing critical insights about LLM limitations in intermediate reasoning. Our results show **clear stratification of model abilities** (Gemini 2.5 Pro 37%, DeepSeek-R1: 25%, o1: 18%, GPT-4o: 7%), demonstrating that this dataset effectively reveals **fundamental differences in reasoning capabilities** that likely transfer to other domains.
>
> **Q2: While the EED Score effectively assesses final answers, it does not verify the correctness of intermediate reasoning steps, a critical aspect of true problem-solving.**
>
> **A2:** Thanks for your detailed comment. We fully agree that verifying intermediate reasoning steps is crucial. Current binary scoring **completely ignores the reasoning process**, providing no insight into whether models achieve correct answers through proper understanding or spurious correlations.
>
> Our EED Score, while not perfect, represents a **significant breakthrough**. Consider calculating a pendulum's period—one can use torque analysis ($mgl\sin\theta = -ml^2\ddot{\theta}$), energy conservation ($\frac{1}{2}ml^2\dot{\theta}^2 + mgl(1-\cos\theta) = \text{Const}$), or work with horizontal displacement. Each approach yields $T = 2\pi\sqrt{l/g}$ but with **entirely different intermediate steps**, illustrating the inherent complexity.
>
> Despite these challenges, EED Score can **partially capture intermediate reasoning**—for instance, verifying all three electromagnetic terms in Equation (5). We acknowledge that physics problems admit **multiple valid solution paths**, making rule-based evaluation nearly impossible, while model-based approaches are unreliable as LLMs struggle with these problems themselves.
>
> Nevertheless, EED Score provides a **systematic attempt** to evaluate whether models follow physically meaningful solution paths, offering **crucial insights previously unavailable**. We view it as an important stepping stone toward more sophisticated reasoning evaluation methods.
>
> We hope these clarifications demonstrate PHYBench's substantial contributions and our commitment to continued development.

---

> > ### Comment · Reviewer_ncFb · 2025-08-06
> >
> > Thanks very much for the authors’ response.
> > I would like to state that their reply answered my questions, but I still keep my score since I am not expertise in this field.
> > Despite that, this paper indeed has a good quality in terms of writing, experiments, and other aspects, which supports its acceptance to some extent.
> > Of course, I also don’t object to the rejection due to some critical issues as pointed by by Reviewer EsZ9.

---

> > > ### Author Response · Authors · 2025-08-06
> > > **Thanks for appreciation, reaffiming core contributions of PHYBench**
> > >
> > > Thank you for your thoughtful feedback and for acknowledging the quality of our work. We greatly appreciate your balanced perspective on both the strengths of our paper and the concerns raised.
> > >
> > > Regarding the expertise question you mentioned, we'd like to clarify that while Reviewer EsZ9 has emphasized their Physics Olympiad Gold Medalist background, the PHYBench team comprises **over 50** Physics Olympiad Gold Medalists from IPHO, APHO, and other prestigious competitions. **We are confident in our team's expertise** and are happy to further address the concerns raised.
> > >
> > > Addressing scalability and complexity concerns: You mentioned potential issues with EED Score's applicability. As detailed in our paper, EED Score has **O(n²)** computational complexity, which is not prohibitive, and supports parallel computation. For PHYBench's evaluation needs, this is not computationally expensive. Regarding complex expressions, we refer to Equation (5)'s electromagnetic formulation as an example—expressions of similar or greater complexity comprise over 30% of PHYBench. Our extensive testing confirms that EED Score handles complex expressions effectively and is specifically designed for expression-based answers.
> > >
> > > Reaffirming PHYBench's core contribution: The fundamental value of PHYBench lies in **being consistently capable of genuinely measuring physics reasoning ability**. Unlike previous works like OlympiadBench and UGPhysics that used publicly available problems (risking data contamination), PHYBench represents original, expert-crafted content. The benchmark's effectiveness is demonstrated by an undeniable fact: despite three months of optimization efforts by major model developers, state-of-the-art models have only improved from 37% to 42% accuracy. This reveals significant gaps in current AI systems' physics understanding.
> > >
> > > Addressing novelty concerns: Through our detailed discussions, we have thoroughly demonstrated PHYBench's novelty. Even when Reviewer EsZ9's experimental proposals remained vague, we provided LLM+search validation proving our problems are not mere copies or narrative adaptations, but genuine physics model-level modifications designed to eliminate ambiguity and evaluation noise. We remain confident that even under more stringent novelty verification requirements, PHYBench's substantial advantages would be evident. **Notably, Reviewer EsZ9 has yet to provide clearer experimental protocols or comparison baselines, which itself speaks to PHYBench's quality.**
> > >
> > > We hope this clarifies our position and core contributions. Thank you again for recognizing our work's contribution, and we look forward to your continued support.

---

### Official Review · Reviewer_6Sgh · 2025-07-01

**Rating:** 4
**Confidence:** 3

**Summary:**

This paper introduces PHYBench, a benchmark with 500 original physics problems spanning a wide range of difficulties from high school to Physics Olympiad level. It aims to address existing limitations in reasoning model evaluations by providing a more challenging and rigorous framework. The authors also introduce the Expression Edit Distance (EED) Score, a novel evaluation metric designed to more effectively assess models' mathematical reasoning abilities, thus providing a finer-grained performance measure. Evaluation of various LLMs on PHYBench shows that even state-of-the-art models significantly underperform compared to human experts, highlighting the complexity of the task.

**Dataset Code Accessibility:**

Yes

**Ethical Considerations:**

Yes, there are ethics concerns that require attention by the authors

**Final Justification:**

After discussion and considering the suggestions of other reviewers, I maintain my score of 4.

**Limitations Weaknesses:**

While the contributions are strong, there are a few areas that could be improved:


EED Score: The EED Score, while an important innovation, could potentially miss intermediate reasoning steps. The paper mentions that future work will focus on expanding the dataset and covering more intermediate reasoning steps, which could further strengthen the evaluation process.

Dataset Expansion: The benchmark is still relatively small (500 problems), which may not be sufficient for more exhaustive model training or testing, especially with emerging models that can handle larger datasets.

**Strengths Contributions:**

The main strength of the work lies in the introduction of PHYBench as a challenging and original evaluation benchmark. By focusing on complex, multi-step physics problems, PHYBench avoids the pitfalls of oversimplified reasoning tasks that dominate many existing benchmarks. Additionally, the paper's design of the EED Score allows for partial credit scoring and nuanced assessments of solution correctness, improving evaluation reliability and sample efficiency. This is a noteworthy contribution, particularly for benchmarks involving symbolic mathematical reasoning.

Furthermore, the paper effectively differentiates models’ reasoning capabilities and highlights the performance gap between LLMs and human experts, providing valuable insights into current LLM limitations in solving physics problems. The use of a human baseline in evaluation is a strong point, offering a clear comparison for model performance.

---

> ### Author Rebuttal · Authors · 2025-07-29
>
> **Response to Reviewer 6Sgh**
>
> We greatly appreciate your recognition of our strong contributions and the constructive feedback provided. We address the two main concerns below:
>
> **Q1: The EED Score, while an important innovation, could potentially miss intermediate reasoning steps. The paper mentions that future work will focus on expanding the dataset and covering more intermediate reasoning steps, which could further strengthen the evaluation process.**
>
> **A1:** Thank you for your thoughtful comment. We fully acknowledge that EED Score cannot capture all nuances of intermediate reasoning, particularly given the multiple valid solution paths in physics problems.
>
> Consider the simple pendulum period calculation as an illustrative example. Starting from torque analysis with angular displacement θ:
>
> $mgl \sin(θ) = -ml² \ddot{θ}$
>
> Using small angle approximation: $\ddot{θ} + (g/l)θ = 0$
>
> Alternatively, through energy conservation:
>
> $\frac{1}{2}ml²\dot{θ}² + mgl(1-\cos(θ)) = \text{Const}$
>
> After approximation and differentiation: $\ddot{θ} + (g/l)θ = 0$
>
> A third approach using horizontal displacement $x$ yields: $\ddot{x} + (g/l)x = 0$
>
> Despite identical final periods $T = 2π\sqrt{l/g}$, these methods involve **completely different intermediate mathematical structures**, highlighting the complexity of intermediate evaluation.
>
> Compared to binary scoring that **entirely ignores reasoning processes**, EED Score represents a **significant breakthrough**. It provides the first systematic attempt to assess whether models follow physically meaningful solution paths—for example, **verifying the presence of all required electromagnetic interaction terms** in complex equations.
>
> While challenges remain (rule-based methods struggle with algebraically different but equivalent forms; model-based evaluation is unreliable when LLMs themselves fail on these problems), EED Score achieves a **crucial balance between evaluation depth and computational feasibility**, marking an important advancement toward more sophisticated reasoning assessment.
>
> **Q2: Dataset Expansion: The benchmark is still relatively small (500 problems), which may not be sufficient for more exhaustive model training or testing, especially with emerging models that can handle larger datasets.**
>
> **A2:** We respectfully but strongly disagree that PHYBench is insufficient for evaluation. While 500 problems may seem modest, our results demonstrate this size **already effectively differentiates model capabilities** (Gemini 2.5 Pro 37%, DeepSeek-R1: 25%, o1: 18%, GPT-4o: 7%). Each problem undergoes our **rigorous multi-round validation process** (Figure 3), ensuring novel, correct, and reliably evaluable content.
>
> PHYBench makes **three breakthrough contributions**:
> 1. The **first human-curated benchmark** specifically designed for rigorous evaluation of complex physical reasoning, with guaranteed problem novelty through our stringent curation pipeline
> 2. The **EED Score** that provides fine-grained, continuous assessment of solution correctness with **204% improved sample efficiency** compared to binary scoring
> 3. **Revealing critical insights** about LLM limitations—models introduce incorrect conditions in intermediate steps and lack human-like self-correction abilities
>
> We prioritize **quality over quantity**—adding more problems would likely reinforce rather than alter the clear performance stratification already demonstrated. We hope the reviewer recognizes these **substantial achievements** in establishing a fair, high-quality benchmark that advances our understanding of LLM reasoning capabilities. We remain committed to thoughtful expansion while maintaining the high standards that make PHYBench a valuable contribution to the field.

---

> > ### Comment · Reviewer_6Sgh · 2025-08-06
> >
> > Thank you for your response. Overall, I hold a positive view of this dataset; its scope and motivation merit acceptance. After considering the other reviewers’ comments, I still have some (non-critical) concerns regarding dataset scale and the quality of the questions, but the work ranks in the upper-middle tier. Therefore, I maintain my positive score of 4.

---

> > > ### Author Response · Authors · 2025-08-06
> > > **Thank you for your recognition and evaluation!**
> > >
> > > We understand you still have some concerns regarding the data scale and the quality of the questions, and we hope to resolve them here. Regarding the data scale, the statistical uncertainty for 500 problems is about 1.5-2%, lower enough to distinguish current models (e.g., Gemini 2.5 Pro at 37%, DeepSeek-R1 at 25%, o1 at 18%, and GPT-4o at 7%). While expanding the dataset to 3000 problems could reduce the statistical uncertainty ($\sigma$) from ~2% to ~1%, this marginal gain is easily overshadowed by errors introduced by evaluation metrics or potential biases from lower-quality data. We therefore chose to prioritize data quality and the accuracy of our evaluation metric. This principle is also seen in benchmarks like GPQA-Diamond (178 problems) and MATH (500 problems), which are now considered by many to be lacking in difficulty, not in size.
> > >
> > > Regarding your comment on a size that may not be "sufficient for more exhaustive model training or testing," we would like to clarify our design intent. Our primary goal for PHYBench is to serve as an evaluation set, not a training set for RL experiments. For the community's "testing" needs during development, we have open-sourced a 100-problem subset to act as an indicator. Given that reasoning through PHYBench's long chains of thought is computationally intensive (requiring hours on 8x A100s for the full set for a 14B model), most developer needing a companion indicator during training would naturally downsample to 100 problems or even fewer.
> > >
> > > Ultimately, we are committed to providing the community with solid conclusions about which models are stronger, for which the 500-item set is both appropriate and sufficient. We are continuously updating our leaderboard as a service and welcome requests to measure your models on the full set.
> > >
> > > Regarding our quality, we strictly control our problem generation pipeline and have meticulously validated the evaluation metric of each problem. However, we believe the most important factor is the effort and enthusiasm from our contributors, which is embodied in every single problem in the dataset. The most important external indicator of this quality, in our view, is the leaderboard's ability to evaluate models, both now and in the future.

---

### Official Review · Reviewer_NtyQ · 2025-07-01

**Rating:** 6
**Confidence:** 4

**Summary:**

The authors build a new dataset called PHYBench, which consists of 500 original curated physics problems for LLM evaluations. It’s shown that most state-of-the-arts LLMs underperform human experts (i.e. Chinese Physics Olympiad gold medalists) in this benchmark under both the standard accuracy score and the more fine-grained Expression Edit Distance (EED) metric proposed by the authors. The PHYBench requires more tokens usage during the reasoning process of LLMs compared to other existing benchmarks like OlympiadBench, MATH500, AIME 2024, and GPQA and are more challenging as reflected in the lower accuracy scores obtained by SOTA LLMs compared to other benchmarks. Different LLMs also have consistent test-time scaling behaviors on PHYBench.

The authors go further and show that most of the errors from LLMs on PHYBench come from the so-called “semantic reasoning” error type, which is the failure to derive equations from physical contexts and problem descriptions rather than interpreting the given physical conditions and manipulating symbolic equations. Most LLMs are also vulnerable to different types of perturbations in the provided partial solution traces and are potentially doing superficial reasoning instead of having genuine physical understanding. Overall, PHYBench has highlighted many shortcomings of current SOTA LLMs in physical reasoning and understanding.

**Dataset Code Accessibility:**

Yes

**Ethical Considerations:**

No, there are no or only very minor ethics concerns

**Final Justification:**

I have looked through other reviews. Some of the main issues raised by other reviewers are 1) small dataset size, 2) appropriateness of the EED Score metric, 3) lack of data leakage prevention measurement, and 4) overclaiming. I believe 1) and 2) are relatively minor issues, and I understand the difficulties of curating novel problems. I do want to mention that I'm trusting the authors in their data curation process since I'm aware of the existence of private problem sets for Physics Olympiad. So take my stance here with a grain of salt. I also encourage the authors to include quantitative measurements of data leakages as suggested by Reviewer EsZ9.

Aside from these issues, I see the most important contributions of this paper as highlighting existing limitations of current LLMs in physical reasoning. The evaluation results of public and proprietary LLMs in this paper have demonstrated this point already. I believe this is an important contribution to the community, and I would certainly like to see how the community reacts to the observations made in this paper in-person at the conference. Therefore I recommend acceptance.

**Limitations Weaknesses:**

- As the authors mentioned, there is a lack of sufficient evaluations of intermediate reasoning chains
- It’s a bit hard to understand the source of the physics problems and why is it not publicly available on the Internet, though I’m not sure if the authors are willing or able to disclose more details on this.
- Question: Do the performances of LLMs on PHYBench vary a lot with different instruction templates? One concern might be that the poor performances can be due to uncommon instruction templates which are slightly out of distribution.

**Strengths Contributions:**

- The originality and the difficulty of the PHYBench dataset is already quite a good contribution to the community. The demonstration of the failures of SOTA LLMs over the PHYBench dataset compared to other potentially data-contaminated benchmark like OlympiadBench, MATH500, AIME 2024, and GPQA is also an important contribution, which highlights a wide gap between LLM reasoning and true physical understanding. I believe this contribution alone is worth a strong acceptance.
- The proposed EED metric is a good contribution and it’s more fine-grained compared to the accuracy score.
- The error types analyses and the perturbation experiments are quite an interesting read and show in what aspects are current LLMs failing.
- The papers are well-written and easy to follow.

---

> ### Author Rebuttal · Authors · 2025-07-29
>
> **Response to Reviewer NtyQ**
>
> We sincerely appreciate your constructive feedback. We are committed to making PHYBench a fair, high-quality, and challenging benchmark that advances the evaluation of physical reasoning in LLMs.
>
> **Q1: As the authors mentioned, there is a lack of sufficient evaluations of intermediate reasoning chains**
>
> **A1:** This is a valid concern. We acknowledge the inherent limitations in evaluating intermediate reasoning steps. Physics problems often admit **multiple valid solution paths**, making it challenging to assess intermediate steps with complete certainty.
>
> For instance, consider calculating the period of a simple pendulum (mass $m$, length $l$, under gravity $g$). Using the angular displacement $\theta$ from vertical as the variable, one can apply torque analysis:
>
> $mgl \sin(\theta) = -ml^2 \ddot{\theta}$
>
> With small angle approximation $\sin(\theta) \approx \theta$, this yields:
>
> $\ddot{\theta} + \frac{g}{l}\theta = 0$
>
> Alternatively, using energy conservation:
>
> $\frac{1}{2}ml^2\dot{\theta}^2 + mgl(1-\cos(\theta)) = \text{Const}$
>
> With $\cos(\theta) \approx 1-\theta^2/2$ and differentiating:
>
> $\ddot{\theta} + \frac{g}{l}\theta = 0$
>
> Or one could use horizontal displacement $x$ as the variable, obtaining:
>
> $\ddot{x} + \frac{g}{l}x = 0$
>
> All three approaches yield the same period $T = 2\pi\sqrt{l/g}$, but their intermediate steps and differential equations differ completely. This illustrates why evaluating intermediate reasoning is inherently complex.
>
> While binary (0/1) scoring completely ignores these reasoning processes, our EED Score represents a significant breakthrough by providing a more nuanced assessment that can distinguish between fundamentally different but equally valid approaches, while maintaining computational feasibility and objectivity.
>
> Additionally, perfect intermediate evaluation remains extremely challenging: rule-based approaches fail due to multiple valid solutions with drastically different algebraic forms; model-based methods are unreliable as current LLMs cannot solve these problems themselves; Process Reward Models are infeasible given limited data and difficulty defining discrete "steps" in continuous derivations. Despite these challenges, EED Score provides valuable insights into reasoning quality—an important first step toward more sophisticated evaluation methods.
>
> **Q2: It's a bit hard to understand the source of the physics problems and why is it not publicly available on the Internet.**
>
> **A2:** Thank you for this detailed comment. Our problems are sourced from non-public resources including internal problems from physics competition training institutions and regional competition selection exams. These materials have not been digitized or published on any websites, forums, or online databases. Importantly, we obtained proper permissions for adaptation and have substantially modified the original problems to ensure originality while preserving their physics content and difficulty level. For undergraduate-level problems, we collaborated with faculty members who provided final exam questions and in-class exercises, all with appropriate permissions.
>
> **Q3: Do the performances of LLMs on PHYBench vary a lot with different instruction templates?**
>
> **A3:** Thank you for this insightful comment. To address this important concern, we conducted additional experiments testing five categorically different instruction templates. Our original template in the paper is:
> ```
> You are a physics expert. Please read the following question and provide a step-by-step solution.
> Put your final answer, which must be a readable LaTeX formula, in a \boxed{} environment.
> Question: {problem from PHYBench}
> Answer:
> ```
> We compared this with four alternative templates:
>
> **Role-based Expert**
> ```
> You are a physics professor. Please solve this problem as you would demonstrate to your students,
> showing clear reasoning and calculations. Final answer in \boxed{}.
> Problem: {problem}
> ```
> **Structured Problem-Solving**
> ```
> Physics Problem: {problem}
> Please solve by:
>
> Identifying given information
> Stating relevant physics principles
> Setting up equations
> Solving step-by-step
> Final answer in \boxed{}
> ```
>
> **Conversational CoT**
> ```
> Let's work through this physics problem together. Think carefully about what physical concepts apply,
> show all your reasoning, and arrive at the answer step by step. Place your final result in \boxed{}.
> Question: {problem}
> ```
> **Direct**
> ```
> Problem: {problem}
> Solution (final answer in \boxed{}):
> ```
> The results are shown below:
>
> | Model | Direct | Professor | Structured | CoT | Original | Max Diff |
> |-------|--------|-----------|------------|-----|----------|----------|
> | gpt-4.1-2025-04-14 | 22.64% | 23.01% | 24.89% | 25.04% | 25.02% | 2.40% |
> | o4-mini | 30.50% | 33.32% | 31.85% | 33.86% | 31.71% | 3.36% |
> | Gemini-2.5-Flash | 43.22% | 38.73% | 41.70% | 42.58% | 42.68% | 4.49% |
>
> The results demonstrate remarkably stable performance across all templates, with maximum variations of only **2.40-4.49%** for each model. All models show similar fluctuation patterns, with slightly better performance on structured/CoT templates but no dramatic differences. This minimal variance confirms that PHYBench genuinely evaluates physical reasoning capabilities rather than sensitivity to prompt engineering, reinforcing the robustness of our benchmark.

---

> > ### Comment · Reviewer_NtyQ · 2025-08-04
> >
> > Thank you for your detailed response. It's also good to see stable performance across a variety of instruction templates. I maintain my score.

---

> > > ### Author Response · Authors · 2025-08-04
> > > **Sincere Thanks to Reviewer NtyQ**
> > >
> > > Thank you for your trust and confidence in our project. We greatly appreciate your recognition and support of PHYBench. We are also pleased to see that PHYBench demonstrates stable measurement results across different prompt templates, which reinforces its reliability as an evaluation benchmark. We believe it will bring valuable contributions to the community. Thank you once again for your acknowledgment!

---

### Decision · Program_Chairs · 2025-09-18

**Decision:**

Accept (poster)

**Comment:**

PHYBench is a valuable benchmark that advances the rigorous evaluation of LLMs in physical reasoning. Its methodological care, diagnostic clarity, and meaningful community contribution justify its inclusion at the conference, even if certain aspects (novelty validation, dataset size) can be improved in future work. The authors' constructive rebuttal, willingness to revise claims, and transparency throughout the discussion phase further support this recommendation.

The review set includes both strong advocates (e.g., NtyQ) and skeptical voices (EsZ9). While the concerns raised by EsZ9 are not entirely resolved, they have been meaningfully addressed and do not appear to undermine the entire contribution.

Strengths:

PHYBench clearly differentiates SOTA models. Despite its public availability, model performance has only improved marginally (e.g., from 37% to 42% over months), underscoring its robustness and difficulty. The proposed EED Score is a meaningful innovation. Problems were manually created or substantially adapted, reviewed through multi-round human vetting, and designed to eliminate scoring ambiguity—ensuring both quality and evaluability. Authors provided substantial clarification, including empirical novelty validation using LLM-based and search-augmented evaluations, template robustness experiments, and concrete acknowledgment of problematic claims (e.g., revising "entirely original" to “originally curated or carefully modified”).

Weaknesses:

Reviewer EsZ9 raised persistent concerns about the novelty level of PHYBench problems. While authors made a strong good-faith effort to apply a structured novelty analysis, some problems likely fall into "low-novelty" categories, and a rigorous, automated validation pipeline is still lacking. Although acknowledged as sufficient for evaluation, the relatively small size (500 problems) raises questions about generalizability and scalability for broader training or longitudinal research use.